# Regret Is Weighted Forgetting*

## Abstract

How much of an agent's regret comes from a bad representation, and how much from a bad policy? Here I give an answer. For a fixed representation $M$ and finite evaluation distribution over history-test pairs, the minimum average normalized regret over all $M$-based policies equals the minimum margin-weighted deletion cost needed to make the optimal bet single-valued on each representation-test cell $(M(h), T)$. A policy-wise decomposition then splits any actual policy's regret into irreducible aliasing cost plus avoidable within-cell misreporting. A Stack-Theoretic reformulation identifies the same quantity as a deficit in weighted weakness on a lifted task constructed from the evaluation support (where weakness is normally the degree to which a policy leaves open unseen diagnostic continuations). I use the identity to derive several corollaries, including a representation-convergence theorem in pure RL language, a regret-based partial order on abstractions, Lipschitz stability of $K_\rho$ under margin estimation error, and connections to free energy and multi-agent coordination. A cross-framework corollary converts the regret floor into a generalisation probability. Under the canonical independent prior, the optimal $M$-based policy generalises with probability $e^{-K_\rho(M)}$. The multi-class generalisation to $K > 2$ diagnostic outcomes is proved. Controlled POMDP experiments confirm the decomposition is numerically exact and that $K_\rho$ discriminates between representations where accuracy and raw impurity do not. The weakness-maximisation theorems predict optimal generalisation through least commitment, but their formal object (the extension of a policy in an embodied language) does not have a direct analogue in neural network function approximation. Bridging that gap is identified as an open problem.

## 1 Introduction

When an agent compresses its observation history into a finite representation, it loses information. Some of that loss is harmless. Some of it makes the correct action ambiguous inside a representation cell, creating irreducible regret that no policy layered on top of the representation can fix. How much regret does a given representation force?

This paper answers with a decomposition. Fix a representation $M$ and quotient history-test pairs by the representation-test cells $(M(h), T)$. The minimum average normalized regret over all $M$-based policies is the minimum margin-weighted forgetting cost $K_\rho(M)$ needed to make the optimal bet single-valued on each such cell (Theorem 11). For any actual $M$-based policy, total regret decomposes into $K_\rho(M)$ plus an avoidable within-cell misreporting term (Theorem 15). This is a representation-quality diagnostic analogous to the bias–variance decomposition in supervised learning: it tells you which part of the error is structural and which part is fixable by better optimisation.

The same quantity has a second life. In the Stack-Theoretic framework for generalisation-optimal learning (Bennett, 2023a; 2025c;d), the natural defect variables are weakness, extension size, and selective forgetting on embodied task extensions. I prove that $K_\rho(M)$ is a weighted weakness deficit on a lifted diagnostic task (Theorem 22). This connects RL regret to the broader Stack-Theoretic programme, which already links generalisation-optimal learning to selective memory, intervention-sensitive causal structure, free energy, and multiscale biological organisation (Bennett, 2024b; 2025c). It also connects Stack Theory to the standard RL

---

*This manuscript was revised with assistance from a large language model used as a writing and editing aid. The ideas, claims, formal statements, proofs, and citation judgments were checked by the authors and remain the authors' responsibility.

and causal world-model literature (Sutton & Barto, 2018; Richens & Everitt, 2024; Nayebi, 2026; Richens et al., 2025). The Stack-Theoretic side of the bridge is not a contribution of this paper as the Enactive General Reinforcement Learning (EGRL) programme, its selective-memory clauses, and the pair-proxy machinery were already theorem-level in Bennett (2024a; 2025a;b;d). EGRL is the Stack Theory's translation of reward-labelled interaction histories into the task-extension language. It uses positive and negative subhistories indexed by a binary reward predicate, ranks admissible policy pairs by a weakness-based EGRL pair proxy, and allows selective forgetting of outputs that contradict otherwise good policies. The present result is a contribution to the mathematical core of that broader programme.

A scope note on the term *regret*. The identity below concerns fixed-distribution normalised betting regret in the sense of Nayebi (2026), namely $\delta_i = 1 - V_i^\pi / V_i^\star$ averaged over a finite evaluation distribution over history-test pairs. It is not cumulative regret, MDP value loss, or worst-case minimax regret. The bridge transports cleanly to those settings only in special cases. Keeping this scope explicit is what allows the cell-by-cell decomposition to be exact rather than up-to-constants.

The identity is also a proof technique. I use it to derive several direct corollaries that illustrate the bridge's utility. A representation-convergence theorem in pure RL language says that any two minimal zero-regret representations must induce the same partition on the informative support (Theorem 25). A regret-based partial order on abstractions complements the existing bisimulation-metric order (Theorem 24). A Lipschitz stability result shows that $K_\rho$ degrades gracefully under margin estimation error (Theorem 26). A cross-framework corollary converts the regret floor into a generalisation probability: under the canonical independent prior, the optimal $M$-based policy generalises with probability $e^{-K_\rho(M)}$ (Theorem 28). Further corollaries connect irreducible regret to a free-energy floor (Theorem 27) and quantify the coordination cost of shared representations in multi-agent settings (Theorem 29). The multi-class generalisation to $K > 2$ diagnostic outcomes is proved as Theorem 18.

The rest of the paper is organized as follows. Section 2 gives related work. Section 3 proves the exact reduction and policy-wise decomposition. Section 4 rewrites the same result in native Stack-Theoretic language. Section 5 derives new results via the bridge. Section 6 gives controlled POMDP experiments confirming the identity and testing $K_\rho$ as a representation diagnostic. Section 7 discusses the prospects for using the bridge to guide representation learning in RL, identifies the central open problem of operationalising formal weakness in continuous function approximation, and summarises preliminary experimental findings.

## 2 Related work

### 2.1 Selection theorems and genealogy

The broad claim that competence pressure forces agents to acquire internal structure has several independent lines. The Stack-Theoretic line proves weakness optimality under the maximally uninformative extension model (2023) and derives intervention-sensitive causal identities under explicit preconditions (2023–2025) (Bennett, 2023a;b; 2025c;d). Richens and Everitt derive robust-causal-model results under distributional shift (2024) (Richens & Everitt, 2024). Nayebi works in a standard POMDP setting and derives quantitative selection theorems for predictive state, memory, modularity, regime tracking, and recoding match (2026) (Nayebi, 2026). These five structural selection theorems are distinctive contributions of Nayebi's programme that do not have direct Stack-Theoretic antecedents. The two frameworks also handle noise differently. Rabin and Scott showed that the languages recognisable by deterministic and non-deterministic finite automata coincide (Rabin & Scott, 1959). In the Stack-Theoretic framework, the role played by stochastic policies in Nayebi's setting is played by selective forgetting, which handles noise and inconsistency by discarding outlying data rather than by randomising over actions (Bennett, 2023b; 2025d). A detailed comparison is in Table 1; a claim-by-claim antecedent inventory is in Appendix C. The regret identity, the policy-wise decomposition, the multi-class generalisation, the measurable version, and the derived consequences in Section 5 are new to this paper.

The representational results sit inside a broader debate. Brooks pushed intelligence without explicit internal representation, while Marr and later work in representation learning and transfer emphasise structured internal states (Brooks, 1991; Marr, 1982; Bengio et al., 2013; Yosinski et al., 2014). The causal representation

Table 1: How the three lines relate.

|  | **Bennett** | **Richens & Everitt** | **Nayebi** |
|---|---|---|---|
| Main setting | Task extensions and policy weakness | Causal models under distributional shifts | POMDP betting tasks and predictive tests |
| Competence variable | Weakness, with selective forgetting for noise | Robust regret-bounded adaptation | Average-case normalised regret |
| Core structural conclusion | Optimal generalisation selects weak policies and, under preconditions, intervention-sensitive causal identities | Robust agents must learn an approximate causal model | Low regret selects predictive state, memory, modularity, regime tracking, and recoding match |

learning programme argues that disentangled causal variables are the natural targets (Schölkopf et al., 2021; Goyal & Bengio, 2022; Ke et al., 2021; Bengio et al., 2020). Nayebi's recoding theorem and Cao and Yamins' contravariance principle are best read against that background (Nayebi, 2026; Cao & Yamins, 2024).

## 2.2 State abstraction and bisimulation

The representation-test quotient in our bridge theorem is a form of state abstraction. The state abstraction literature in RL is mature and provides important context. Li, Walsh, and Littman give a unified treatment of five abstraction schemes for MDPs, ranging from model-irrelevance to $\pi^*$-irrelevance and bisimulation, and analyse which preserve optimal planning (Li et al., 2006). Givan, Dean, and Greig earlier studied equivalence notions and model minimization for MDPs, identifying bisimulation as the finest useful equivalence (Givan et al., 2003). Ferns, Panangaden, and Precup introduced quantitative bisimulation metrics that measure state similarity continuously rather than through hard partitions, providing value-function bounds on the cost of aggregation (Ferns et al., 2004; 2011). Abel and collaborators extended these ideas to approximate and lifelong settings (Abel et al., 2016; 2018).

Our quotient cells differ from bisimulation cells in an important respect. Bisimulation groups states by identical transition and reward structure. Our representation-test cells group history-test points by what the representation $M$ can see and which diagnostic test is applied. The bridge theorem then says how much regret this grouping forces. In that sense it complements the bisimulation programme: bisimulation metrics bound value-function loss, while our theorem identifies exact regret cost at a given quotient.

More recent deep RL work brings bisimulation ideas into the function-approximation regime. Zhang et al. learn representations that respect a bisimulation distance (Zhang et al., 2021), Castro scales bisimulation computation to large deterministic MDPs (Castro, 2020), and Gelada et al. learn latent-space models that preserve MDP structure (Gelada et al., 2019). Nayebi's selection theorems and our bridge result operate at a more abstract level. They characterise when low regret forces a representation to preserve certain distinctions, regardless of how the representation is learned.

*Remark* 1 (Bisimulation distance and cell impurity). In an MDP with known transition and reward structure, if two states $s, s'$ belong to the same representation-test cell $C$ and the bisimulation metric $d_\sim(s, s')$ of Ferns et al. (2004) is large, then $s$ and $s'$ are likely to carry different optimal actions, so their cell will be impure and will contribute positively to $K_\rho(M)$. Conversely, if every pair in a cell has $d_\sim(s, s') = 0$, then the cell is pure and contributes zero. So $K_\rho(M)$ can be read as an aggregate measure of how much bisimulation-relevant structure the representation has thrown away. The two formulations are complementary: bisimulation metrics bound value-function loss, while $K_\rho(M)$ gives exact regret cost at a given quotient.

## 2.3   Predictive state and world models

A parallel line of work asks what internal structure low-regret agents must have by reasoning about predictive sufficiency. Littman, Sutton, and Singh introduced predictive state representations, showing that observable predictions can serve as a complete state description without latent-variable modeling (Littman et al., 2001). Singh, James, and Rudary extended the framework to controlled systems (Singh et al., 2004), and Boots, Siddiqi, and Gordon connected PSR learning and planning (Boots et al., 2011). The approximate information state literature addresses partial observability more directly by identifying sufficient statistics for near-optimal planning (Subramanian et al., 2022).

On the model-learning side, Ha and Schmidhuber's world models (Ha & Schmidhuber, 2018), Hafner et al.'s Dreamer agent (Hafner et al., 2020), and Schrittwieser et al.'s MuZero (Schrittwieser et al., 2020) demonstrate that learned latent dynamics can support competitive planning. Richens, Everitt, and Abel give a theoretical counterpart, proving that general agents need world models under distributional shift (Richens et al., 2025). Our bridge theorem gives a different angle on the same family of questions. Rather than asking what structure a model must have, it asks how much regret a fixed representation forces, measured in weighted forgetting terms.

## 2.4   Causal inference and the causal hierarchy

The causal side of the genealogy involves more than Richens and Everitt alone. Pearl's structural causal model framework (Pearl, 1995; 2009) and the Spirtes–Glymour–Scheines algorithmic tradition (Spirtes et al., 2000) provide the language in which "learning a causal model" is made precise. Bareinboim et al.'s causal hierarchy theorem formalises the separation between observational, interventional, and counterfactual reasoning (Bareinboim et al., 2022). Lattimore, Lattimore, and Reid study causal bandits, where the agent must learn which interventions are effective (Lattimore et al., 2016).

Bennett's emergent causality result (Bennett, 2023b) proved that under weakness maximisation, representations of causal interventions emerge as variables in the embodied language, without presupposing a do-operator. The key argument is that an agent which confuses intervention with passive observation will adopt a more committal (and therefore weaker in the weakness sense) policy than one which distinguishes them, so weakness pressure forces the distinction to emerge. That result, and the subsequent causal identity theorems in (Bennett, 2025c) and (Bennett, 2025d), are antecedent to the Richens-Everitt result that robust agents must learn approximate causal models under distributional shift (Richens & Everitt, 2024).

The two lines are somewhat complementary. Bennett's argument was that since weakness maximisation is the optimal choice of learning heuristic (Bennett, 2025c), an agent that seeks to be optimal within the constraints of its body must use it. Such agents *must* learn causal models (Bennett, 2023b; 2025c). In contrast, Richens and Everitt claimed *robust* agents must learn causal models, basing their claim on reinforcement learning. These are approximately the same claim derived from two different formalisms, which suggests a bridge between Stack Theory and reinforcement learning may be beneficial. In this paper, the regret ordering (Theorem 24) gives a quantitative counterpart to the qualitative claim that robust agents must learn causal models. If the causal partition refines the diagnostic partition $\mathcal{P}_+^\star$, it sits at zero $K_\rho$ and incurs no irreducible regret. If it does not, the bridge tells you how much regret that costs. Richens and Everitt handle worst-case robustness across distributions. The bridge presented here handles the cost at a fixed evaluation distribution.

# 3   Regret as weighted forgetting

Here the bridge is given in finite form. The finite version is the natural one for Stack Theory because the original framework works over finite embodied vocabularies. A measurable version is given in Appendix A. The finite theorem is stated first because it keeps the reduction transparent and matches the original finite task-extension setting.

$$(\mathcal{X}, \mu, y, \rho) \xrightarrow{\text{quotient by } M} \mathcal{C}_M \xrightarrow{\text{drop the cheaper label in each mixed cell}} K_\rho(M)$$

$$K_\rho(M) \xrightarrow{\text{lift to a diagnostic Stack task}} Z - W^\star(M).$$

Figure 1: The reduction at a glance. Quotient history-test points by the representation-test cells induced by $M$. Drop the cheaper label class in each mixed cell. Read the same number as a weighted weakness deficit on the lifted task.

## 3.1 Setup

This subsection defines, in plain English first and then symbolically, the four objects that the rest of the paper assumes. Readers familiar with Nayebi's betting framework can skip to Theorem 5.

**Definition 2** (History, test, conditional). A *history $h$* is the agent's observed interaction record up to a decision point. The space of histories is left abstract. Concrete instantiations include observation-action sequences in a POMDP, sensor readings up to a query time, or an enactive vocabulary statement summarising the body's state. A *test $T \in \mathcal{T}$* is a binary diagnostic query. Its outcome is a binary random variable $Z_T \in \{0, 1\}$. The *conditional*

$$p_T(h) := \Pr(Z_T = 1 \mid h) \in [0, 1]$$

is the true environment probability that the test fires given the history. The agent does not observe $p_T(h)$ directly. The exact identity below uses $p_T(h)$ as a population quantity. Theorem 26 controls what happens when $p_T(h)$ is replaced by an empirical estimate. The multi-class case replaces $Z_T$ by a finite-valued outcome variable and is recovered in Theorem 18.

**Definition 3** (Evaluation distribution). A finite *evaluation distribution* is a finite list of history-test pairs $\{x_i = (h_i, T_i)\}_{i=1}^n$ together with positive weights $\mu_i > 0$ summing to one. The pairs are the support points at which the representation will be diagnosed. The weights say how often each support point is visited under the evaluation regime. They are part of the diagnostic problem, not learned by the agent.

**Definition 4** ($M$-based policy). A *representation $M$* is any function on histories. Its observable output at history $h$ is $M(h)$. An *$M$-based policy* chooses one report probability $q_C \in [0, 1]$ for each cell $C \in \mathcal{C}_M$ where $\mathcal{C}_M$ is the set of equivalence classes of support points under the relation $i \sim_M j$ iff $M(h_i) = M(h_j)$ and $T_i = T_j$. The policy sees only the cell label, not the underlying history. This is what *representation-based* means in this paper.

With these objects fixed, the following compact notation is used throughout. Let

$$\mathcal{X} = \{x_1, \ldots, x_n\}, \qquad x_i = (h_i, T_i),$$

with masses $\mu_i > 0$ and $\sum_i \mu_i = 1$. For each $x_i$, let

$$p_i := p_{T_i}(h_i), \qquad m_i := \left| p_i - \tfrac{1}{2} \right|.$$

Let $y_i \in \{0, 1\}$ denote the optimal bet, where $y_i = 1$ when $p_i \geq \frac{1}{2}$ and $y_i = 0$ otherwise. When $p_i = \frac{1}{2}$, the choice of $y_i$ is arbitrary because the weight below is then zero.

For any policy $\pi$, write $\delta_i(\pi) := 1 - V_i^\pi / V_i^\star$ for the normalised regret on support point $x_i$, where $V_i^\star = \max\{p_i, 1 - p_i\}$ is the optimal success probability and $V_i^\pi$ is the success probability under $\pi$. This is a standard quantity for analysing representation quality in partially observed decision problems (Kaelbling et al., 1998; Sutton & Barto, 2018; Li et al., 2006).

**Definition 5** (Margin weight). For each support point $x_i$, define

$$\rho_i := \frac{2m_i}{\frac{1}{2} + m_i} \in [0, 1].$$

Note that $\frac{1}{2} + m_i = \max\{p_i, 1 - p_i\}$, so $\rho_i$ is the normalised regret penalty per unit of wrong-action mass on test $i$. This is the multiplier that converts wrong-action probability into the normalised regret $\delta_i = 1 - V_i^\pi / V_i^\star$

used in the RL selection-theorem literature (Nayebi, 2026), and the same multiplier arises as the natural prior weight $w_P$ in the Stack-Theoretic programme (Bennett, 2025d). When a test is nearly a coin flip, $\rho_i$ is near zero. When a test is decisive, $\rho_i$ is large. The role is analogous to the margin-based weighting that appears in information-theoretic accounts of lossy compression: uninformative distinctions are cheap to lose, informative ones are expensive (Tishby et al., 2000).

*Remark* 6 (Why the bridge is natural). The quotient cells come from fixing $M$ as in the memory setting, and keeping the test variable visible. The deletion move is selective forgetting on the diagnostic support. The margin weights $\rho_i$, the quotient cells, and the deletion move all have native Stack-Theoretic antecedents in $w_P$, selective forgetting, and the diagnostic extension model (Bennett, 2025d). The prior weight $w_P$ appeared in the Stack-Theoretic generalisation-probability theorems (Bennett, 2023a; 2025d). The margin weight $\rho_i$ appeared later as Nayebi's normalised regret multiplier (Nayebi, 2026). The contribution of this paper is the precise identification of these as the same quantity. The cellwise structure that allows for a precise bridge follows from that identification. Nayebi's betting framework provides the clean RL-side formalisation that gives the bridge its second endpoint.

*Remark* 7 (Scope of the normalisation). The exact identity depends on normalised regret $\delta_i = 1 - V_i^\pi / V_i^\star$ and the resulting margin weight $\rho_i$. Under unnormalised regret $V_i^\star - V_i^\pi$ or under alternative loss functions, the cellwise structure survives but the exact identification requires a different weight. Specifically, unnormalised regret replaces $\rho_i$ with the weight $2m_i$, giving a different margin-weighted deletion cost on the diagnostic support. The bridge to weighted weakness (Theorem 22) requires the normalised form because it matches the prior-weighted quantity $w_P$ already native to Stack Theory.

**Lemma 8** (Pointwise regret decomposition). *Let $\pi$ be any $M$-based policy, and let $q_C$ be its report probability on cell $C$. For $i \in C$,*

$$\delta_i(\pi) = \rho_i\Big((1 - q_C)\mathbf{1}\{y_i = 1\} + q_C\mathbf{1}\{y_i = 0\}\Big).$$

*Proof.* If $y_i = 1$, then $p_i \geq \frac{1}{2}$ and the optimal report is the left bet. The success probability under report probability $q_C$ is

$$V_i^\pi = q_C p_i + (1 - q_C)(1 - p_i),$$

while the optimal success probability is

$$V_i^\star = p_i = \tfrac{1}{2} + m_i.$$

Hence

$$\delta_i(\pi) = 1 - \frac{V_i^\pi}{V_i^\star} = \frac{2m_i(1 - q_C)}{\frac{1}{2} + m_i} = \rho_i(1 - q_C).$$

If $y_i = 0$, the symmetric calculation gives $\delta_i(\pi) = \rho_i q_C$. $\qquad\square$

**Intuition.** Regret is just mistake probability scaled by how informative the test is. That scaling is the only reason the bridge needs a weighted, rather than raw, forgetting variable.

**Definition 9** (Margin-weighted forgetting cost). The margin-weighted forgetting cost[1] of $M$ is

$$K_\rho(M) := \min\Big\{ \sum_{i \in B} \mu_i \rho_i \ \Big| \ B \subseteq \{1, \ldots, n\}, \text{ and for every } C \in \mathcal{C}_M, \ y \text{ is constant on } C \setminus B\Big\}.$$

The cost of forgetting point $i$ is its evaluation mass times its margin weight. You forget just enough so that, inside every representation-test cell, the correct bet becomes single-valued. This is a weighted version of the selective deletion that appears in the state abstraction literature when one asks how much structure a given partition can preserve (Li et al., 2006; Givan et al., 2003).

---

[1] The word *forgetting* here refers to selective deletion of support points to purify representation cells, in the sense of Bennett (2025d). It is not catastrophic forgetting in the continual-learning sense (loss of past task performance under sequential training) and does not refer to weight overwriting in neural networks.

Before stating the theorem, here is a minimal worked example.

**Example 10** (A two-cell warm-up). Take three support points $x_1, x_2, x_3$ under a single test $T$, with $\mu_i = 1/3$ each and conditional probabilities $p_1 = 0.9$, $p_2 = 0.1$, $p_3 = 0.7$. The optimal bets are $y_1 = 1$, $y_2 = 0$, $y_3 = 1$. The margin weights are $\rho_1 = 8/9$, $\rho_2 = 8/9$, $\rho_3 = 4/7$.

Suppose representation $M$ collapses $h_1$ and $h_2$ to the same code while keeping $h_3$ separate. Then $\mathcal{C}_M$ has two cells, $C_{12} = \{x_1, x_2\}$ and $C_3 = \{x_3\}$. On $C_{12}$, $A_{C_{12}} = \mu_1 \rho_1 = 8/27$ (the mass on $y_i = 1$) and $B_{C_{12}} = \mu_2 \rho_2 = 8/27$ (the mass on $y_i = 0$). The minimum is $8/27$, so this cell forces a $K_\rho$ contribution of $8/27$ regardless of which label class the policy commits to. The cell is balanced. On $C_3$, $A_{C_3} = \mu_3 \rho_3 = 4/21$ and $B_{C_3} = 0$, so the minimum is zero. Total $K_\rho(M) = 8/27 \approx 0.296$.

A different representation $M'$ that kept $h_1$ separate from $h_2$ would induce three pure cells, giving $K_\rho(M') = 0$. So aliasing $h_1$ and $h_2$ costs $8/27$ in irreducible regret. The theorem below says the same picture works for arbitrary representations, that the cellwise rule is a theorem rather than a heuristic, and that the same number equals the minimum margin-weighted deletion needed to purify the cells.

**Theorem 11** (Exact reduction). *For every fixed memory representation $M$,*

$$\inf_{\pi \ is \ M\text{-}based} \sum_{i=1}^{n} \mu_i \, \delta_i(\pi) = K_\rho(M).$$

*Proof.* For each cell $C \in \mathcal{C}_M$, define

$$A_C := \sum_{i \in C, \ y_i = 1} \mu_i \rho_i, \qquad B_C := \sum_{i \in C, \ y_i = 0} \mu_i \rho_i.$$

By Theorem 8, any $M$-based policy with report probability $q_C$ on cell $C$ contributes

$$A_C(1 - q_C) + B_C q_C$$

to expected regret on that cell. This is affine in $q_C$, so its minimum over $q_C \in [0, 1]$ is

$$\min\{A_C, B_C\}.$$

Summing over cells gives

$$\inf_{\pi \ is \ M\text{-based}} \sum_{i=1}^{n} \mu_i \, \delta_i(\pi) = \sum_{C \in \mathcal{C}_M} \min\{A_C, B_C\}.$$

Now compute the forgetting cost. To make $y$ single-valued on a fixed cell $C$, one must delete either every $y_i = 1$ point in $C$ or every $y_i = 0$ point in $C$. Deleting anything less leaves both labels in the cell. So the minimum forgetting cost on cell $C$ is

$$\min\{A_C, B_C\}.$$

Summing over cells gives

$$K_\rho(M) = \sum_{C \in \mathcal{C}_M} \min\{A_C, B_C\}.$$

Comparing the two displays proves the theorem. □

*Remark* 12 (Computational cost). Despite the combinatorial phrasing of Definition 9, $K_\rho(M)$ is computable in $O(n)$ time once the representation-test cells are formed, because the minimum deletion on each cell decomposes independently into $\min\{A_C, B_C\}$. No search over subsets is required.

**Intuition.**   Once the task is quotiented by representation-test cell, regret is the cheapest weighted deletion needed to make the correct action a function of that cell. The two quantities are not merely analogous. They are equal.

*Remark* 13 (The optimal bet is selective forgetting). The identification is not merely numerical. The optimal $M$-based policy's per-cell binary choice (bet on $y = 1$ or $y = 0$) is itself a deletion operator. In each cell, the policy commits to one label class and discards the other. The discarded class is the cheaper one, and its margin-weighted mass is the cell's contribution to $K_\rho(M)$. So the RL-side optimal policy is not merely computing a cost equal to selective forgetting. It is *performing* selective forgetting on the diagnostic support.

*Remark* 14 (Relation to weighted Bayes error). The cellwise quantity $\min\{A_C, B_C\}$ is recognisable as a weighted Bayes error on the representation-test partition. From a statistical decision theory perspective, this is the conditional Bayes risk of the partition under a margin-weighted 0-1 loss. The contribution is the three-way identification. It equals Nayebi's normalised regret exactly, not merely up to bounds. It equals the minimum margin-weighted deletion cost in the selective-forgetting sense. And it equals a weighted weakness deficit on a lifted Stack-Theoretic task (Theorem 22). The first gives it RL semantics, the second gives it a forgetting and compression interpretation, and the third connects it to the generalisation-optimal learning programme.

**Corollary 15** (Policy-wise decomposition). *For each cell $C \in \mathcal{C}_M$, choose any*

$$q_C^\star \in \arg\min_{q \in [0,1]} \big(A_C(1-q) + B_C q\big).$$

*Then every $M$-based policy $\pi$ satisfies*

$$\sum_{i=1}^{n} \mu_i \, \delta_i(\pi) = K_\rho(M) + \sum_{C \in \mathcal{C}_M} |A_C - B_C| \, |q_C - q_C^\star|.$$

*In particular,*

$$K_\rho(M) \leq \sum_{i=1}^{n} \mu_i \, \delta_i(\pi),$$

*with equality when the policy is cellwise optimal on every non-tied cell.*

*Proof.* If $A_C > B_C$, then the minimum is attained at $q_C^\star = 1$ and

$$A_C(1 - q_C) + B_C q_C = B_C + (A_C - B_C)(1 - q_C)$$
$$= \min\{A_C, B_C\} + |A_C - B_C| \, |q_C - q_C^\star|.$$

If $A_C < B_C$, the minimum is attained at $q_C^\star = 0$ and the same identity becomes

$$A_C(1 - q_C) + B_C q_C = A_C + (B_C - A_C)q_C$$
$$= \min\{A_C, B_C\} + |A_C - B_C| \, |q_C - q_C^\star|.$$

If $A_C = B_C$, both sides reduce to $A_C$ for every $q_C$. Summing over cells and using Theorem 11 proves the result. $\square$

**Intuition.** This separates two kinds of regret. The first part is representation cost. It is the aliasing penalty you cannot remove without changing $M$. The second part is execution cost. It is the avoidable penalty from betting suboptimally even after $M$ is fixed. That decomposition is useful for the same reason the bias–variance decomposition is useful in supervised learning: it tells you which part of the error is structural and which part is fixable by better optimisation.

**Example 16** (A tiny quotient task). Suppose one mixed cell $C_1$ has $A_{C_1} = 0.30$ and $B_{C_1} = 0.20$, while a second cell $C_2$ is pure with $A_{C_2} = 0$ and $B_{C_2} = 0.25$. Then

$$K_\rho(M) = \min\{0.30, 0.20\} + \min\{0, 0.25\} = 0.20.$$

If an actual policy uses $q_{C_1} = 0.8$ and $q_{C_2} = 0$, then its regret is

$$0.30(1 - 0.8) + 0.20(0.8) + 0.25(0) = 0.22.$$

The extra 0.02 is

$$|0.30 - 0.20| \, |0.8 - 1| = 0.02.$$

The 0.20 is the irreducible cost of aliasing opposite bets inside $C_1$. The extra 0.02 is not representational at all. It is just the cost of choosing the wrong within-cell report.

**Corollary 17** (Unweighted forgetting on the informative region)**.** *In practice one may wish to ignore near-tie diagnostic tests whose margin $m_i$ is close to zero, for instance when empirical estimates of $p_i$ are noisy and tests near $\frac{1}{2}$ carry little signal. The following shows that restricting to sufficiently informative tests recovers an unweighted forgetting picture up to fixed constants.*

*Fix $\gamma \in (0, \frac{1}{2}]$ and let*

$$\mathcal{X}_\gamma := \{i : m_i \geq \gamma\}.$$

*Define the unweighted informative-region forgetting cost*

$$K_\gamma(M) := \min\Big\{ \sum_{i \in B \cap \mathcal{X}_\gamma} \mu_i$$

$$\Big| \; B \subseteq \{1, \ldots, n\}, \; \text{and for every } C \in \mathcal{C}_M,$$

$$y \text{ is constant on } (C \cap \mathcal{X}_\gamma) \setminus B \Big\}.$$

*Then, with $c(\gamma) = \frac{4\gamma}{1+2\gamma}$,*

$$c(\gamma) K_\gamma(M) \leq \inf_{\pi \text{ is } M\text{-based}} \sum_{i \in \mathcal{X}_\gamma} \mu_i \, \delta_i(\pi) \leq K_\gamma(M).$$

*Proof.* On $\mathcal{X}_\gamma$ we have $\rho_i \in [c(\gamma), 1]$. Apply Theorem 11 on the restricted support $\mathcal{X}_\gamma$. The resulting weighted deletion cost lies between $c(\gamma)$ times the unweighted deletion mass and the unweighted deletion mass itself. $\square$

**Intuition.** This recovers the intuitive raw-forgetting picture. If you only look at tests that matter by at least $\gamma$, then weighted forgetting and ordinary forgetting differ only by fixed constants.

## 3.2 Multi-class generalisation

The binary case connects directly to Nayebi's betting setup, but the algebraic structure survives for any finite number of outcome classes.

**Theorem 18** (Multi-class exact reduction)**.** *Fix a finite evaluation support as before, but let $y_i \in \{1, \ldots, K\}$ for $K \geq 2$ outcome classes. For each cell $C \in \mathcal{C}_M$ and class $k$, define*

$$A_C^{(k)} := \sum_{i \in C, \, y_i = k} \mu_i \rho_i.$$

*Then*

$$\inf_{\pi \text{ is } M\text{-based}} \sum_{i=1}^n \mu_i \, \delta_i(\pi) = \sum_{C \in \mathcal{C}_M} \Big( \sum_{k=1}^K A_C^{(k)} - \max_k A_C^{(k)} \Big).$$

*The right-hand side is the minimum margin-weighted deletion cost needed to make $y$ single-valued on each representation-test cell, and it equals the cellwise weighted Bayes error.*

*Proof.* An $M$-based policy assigns a distribution $q_C \in \Delta^{K-1}$ over the $K$ classes for each cell $C$. Its expected regret on cell $C$ is

$$\sum_{k=1}^K A_C^{(k)} (1 - q_C^{(k)}) = \Big( \sum_k A_C^{(k)} \Big) - \sum_k A_C^{(k)} q_C^{(k)}.$$

To minimise this over $q_C \in \Delta^{K-1}$, one must maximise $\sum_k A_C^{(k)} q_C^{(k)}$, which is achieved by putting all mass on the class with the largest $A_C^{(k)}$. So the minimum per-cell regret is $\sum_k A_C^{(k)} - \max_k A_C^{(k)}$. To make $y$ single-valued on $C$, one must delete every point not in the dominant class, costing $\sum_k A_C^{(k)} - \max_k A_C^{(k)}$. Summing over cells gives both sides. $\square$

Table 2: Notation mapping between the Stack-Theoretic and RL sides.

| Stack-Theoretic side | RL side |
|---|---|
| Truth set of a diagnostic statement | Representation-test cell $C$ |
| Child policy $\vartheta$ | Policy $\pi$ |
| Label class selection per cell | Report probability $q_C$ |
| Weight not retained: $w_i \mathbf{1}\{i \notin \mathrm{Ext}(\vartheta)\}$ | Normalised regret $\delta_i$ |
| Weakness deficit $Z - W^\star(M)$ | Min regret $K_\rho(M)$ |
| Weakness-maximising admissible policy | Optimal policy per cell |

## 4   A Stack-Theoretic reformulation

The previous section gives the exact bridge in Nayebi's language. This can now be translated into Bennett's language. The finite diagnostic support becomes the unseen region of a lifted task, and margin weights become a weighted weakness score on the same representation-test quotient.

### 4.1   Notation and prerequisites

This subsection gives self-contained definitions sufficient to verify the Stack-Theoretic form of the bridge (Theorem 22). The reader familiar with Stack Theory may skip ahead.

A *lifted diagnostic task* is defined from the finite evaluation support as follows. Construct a set $U = \{u_1, \ldots, u_n\}$ of *unseen outputs*, one for each support point $x_i = (h_i, T_i)$. Each unseen output $u_i$ inherits the label $y_i \in \{0, 1\}$ and weight $w_i := \mu_i \rho_i$ from its corresponding support point. A *policy on the lifted task* selects, for each representation-test cell $C \in \mathcal{C}_M$, which label class to retain. Its *extension*

$$\mathrm{Ext}(\vartheta) \cap U := \{u_i \in U : y_i \text{ equals the label class retained in cell } C \ni i\}$$

is the set of unseen outputs whose label matches the policy's selection. The policy is *admissible* if it retains outputs from at most one label class per cell.

This vocabulary is standard in the Stack-Theoretic framework (Bennett, 2023a; 2025d) but is used here only in the restricted form above. Table 2 gives the correspondence.

### 4.2   Weighted weakness and the bridge

**Definition 19** (Prior-weighted weakness). Let $U = \{u_1, \ldots, u_n\}$ be a finite unseen region with positive weights $w_1, \ldots, w_n$. For any policy $\vartheta$, let $\mathrm{Ext}(\vartheta)$ denote its extension, i.e. the set of all outputs it is compatible with in the host language (Bennett, 2023a; 2025d). Define its prior-weighted weakness on $U$ by

$$W(\vartheta) := \sum_{u_i \in \mathrm{Ext}(\vartheta) \cap U} w_i.$$

Ordinary weakness counts how many unseen continuations a policy leaves open. Weighted weakness counts how much weighted unseen future it leaves open. The idea is analogous to the information bottleneck: discard what is cheap, keep what is expensive (Tishby et al., 2000).

*Remark* 20 (This is not a new Stack-Theoretic primitive). Bennett (2025c) already defines the more general quantity

$$w_P(\pi) = P\big(S \subseteq \mathrm{Ext}(\pi) \cap U\big)$$

for an arbitrary prior $P$ over subsets $S \subseteq U$. The score $W(\vartheta)$ above is just the finite product-prior specialisation that becomes convenient after taking logs (Bennett, 2025d). So this section is a translation of existing Stack-Theoretic machinery, not a replacement for it.

**Proposition 21** (Independent nonuniform priors give weighted weakness). *Assume each unseen output $u_i \in U$ becomes relevant independently with probability $r_i \in (0,1)$. Then, for any correct child policy $\vartheta$,*

$$\log \Pr(\vartheta \text{ generalises}) = \sum_{u_i \notin \text{Ext}(\vartheta) \cap U} \log(1 - r_i).$$

*Equivalently, maximising generalisation probability is the same as maximising*

$$\sum_{u_i \in \text{Ext}(\vartheta) \cap U} \big(-\log(1 - r_i)\big).$$

*So the Bayes-optimal rule is weighted weakness maximisation with weights*

$$w_i := -\log(1 - r_i).$$

*Proof.* Generalisation occurs when every unseen output omitted by $\vartheta$ fails to become relevant. Independence gives the product

$$\Pr(\vartheta \text{ generalises}) = \prod_{u_i \notin \text{Ext}(\vartheta) \cap U} (1 - r_i).$$

Taking logs yields the display. The remaining term depends on $\vartheta$ only through the retained weighted support. $\square$

This is the weighted version of Bennett's earlier counting theorem. Uniform priors recover ordinary weakness. Biased priors recover weighted weakness. The uniform case is the no-free-lunch baseline: if you know nothing about which unseen outputs matter, you keep as many open as possible (Wolpert & Macready, 1997).

**Corollary 22** (Exact Stack-Theoretic form of the bridge). *Let $w_i := \mu_i \rho_i$. Build a lifted diagnostic task whose unseen outputs are $u_1, \ldots, u_n$, one for each support point $x_i$. Restrict admissible policies so that, on each representation-test cell $C \in \mathcal{C}_M$, the policy retains outputs from at most one label class. Let*

$$Z := \sum_{i=1}^{n} w_i, \qquad W^\star(M) := \max_{\vartheta \ admissible} W(\vartheta).$$

*Then*

$$\inf_{\pi \ is \ M\text{-}based} \sum_{i=1}^{n} \mu_i \, \delta_i(\pi) = Z - W^\star(M).$$

*Moreover, choosing independent prior probabilities*

$$r_i := 1 - e^{-w_i}$$

*turns $W^\star(M)$ into the Bayes-optimal weighted weakness score of the lifted task.*

*Proof.* On the lifted task, retaining a support point $u_i$ means not forgetting the corresponding diagnostic point $x_i$. Admissibility means each cell keeps at most one label class. So maximising retained weighted mass is the same optimisation problem as minimising deleted weighted mass. Hence

$$W^\star(M) = Z - K_\rho(M).$$

Now apply Theorem 11. The final statement follows from Theorem 21 and the choice $r_i = 1 - e^{-w_i}$. $\square$

**Intuition.** Regret is the weighted unseen future that the representation forced you to discard. The earlier counting theorem says generalisation is about how much unseen future you leave open. The bridge says regret is the part of that weighted future lost to aliasing.

*Remark* 23 (Weighted weakness is not generalisation-optimal in the unconditional sense). The Sufficiency and Necessity theorems for plain weakness (Bennett, 2025c;d) are unconditional generalisation-optimality results. Under the maximally uninformative prior over which unseen continuations may become demanded, the weakest correct policy maximises generalisation probability. No weighting is required.

Theorem 21 is a different beast. It says that if you adopt independent relevance probabilities $r_i$ for the unseen outputs, then the rule that maximises generalisation probability under that adopted prior is weighted weakness maximisation with weights $w_i = -\log(1 - r_i)$. The optimality is conditional on the prior. Adopting any non-uniform prior is itself a commitment about which unseen futures matter, so weighted weakness can specialise the agent to that prior at the cost of generalisation under any other.

This is not to say weakness as a *score* is encoding-invariant. The cardinality $|\text{Ext}(\pi)|$ is the number of completions of $\pi$ inside the embodied language $L_{\mathfrak{v}}$, and $L_{\mathfrak{v}}$ depends on the chosen vocabulary $\mathfrak{v}$. Re-encode the agent into a different vocabulary that picks out the same admissible policies, and the cardinality changes. What *is* encoding-invariant is the *optimality* of the weakness rule. The proof of the Sufficiency theorem yields, inside any finite vocabulary $\mathfrak{v}$, the identity $\Pr(\pi \in \Pi_\omega \mid \alpha) = 2^{|\text{Ext}(\pi) \cap U|}/2^{|U|}$, and this is monotone in $|\text{Ext}(\pi) \cap U|$ inside that vocabulary. So in every $\mathfrak{v}$ the prescription "pick the weakest correct policy" is generalisation-optimal under maximal ignorance. The numbers vary across encodings, but the rule does not.

Alternative rules do not have that property. The simplicity rule "pick the shortest correct policy" is provably non-optimal in some vocabularies because the shortest correct policy can fail to be weakness-maximal there (Bennett, 2025c;d, Propositions on simplicity sub-optimality and subjectivity of description length). Weighted weakness "maximise $\sum_{u_i \in \text{Ext}(\vartheta) \cap U} w_i$" may be optimal under a specific prior $P$, but the prior is pinned to the unseen outputs $u_i$ of a particular language. Re-encode the agent and the unseen region changes, the weights have to be re-specified for the new $u_i$, and whether the new weighted rule is optimal in the new language depends on whether the new prior matches the new environment. Plain weakness's optimality survives encoding changes. Alternatives' effectiveness does not.

This paper uses the weighted form because it is necessary for an exact bridge to RL. The RL side already carries the weights $\rho_i$ via normalised regret. The weighted weakness is merely an acknowledgement of what the RL side already commits to. It is not promoted here as a better generalisation criterion than plain weakness, because it does not preserve the encoding-invariance of the optimality result.

### 4.3 Continuity with the earlier EGRL and pair-proxy programme

EGRL here means the "enactive general reinforcement learning" translation from reward-labelled interaction histories into the task-extension language. An EGRL pair proxy is the rule used to rank the admissible positive-negative policy pairs extracted from that history. The weakness EGRL pair proxy ranks such pairs by the size of their joint extension. Prior work constructs instantiated history tasks, reward predicates, admissible pair policies, selective memory for inconsistent histories, and EGRL-style translations from interaction histories into the task-extension language (Bennett, 2024a; 2025a;b;d). The present theorem provides a mathematical core for that broader programme. It fixes a diagnostic goal family, quotients the history support by representation-test cell, and proves that the resulting defect variable is weighted selective forgetting.

**Intuition.** The earlier programme translated the whole interaction loop. This paper isolates the reusable kernel, which is the defect variable at the representation-test quotient.

## 5 Consequences of the bridge

The identity is a proof technique, not just a dictionary. This section derives results via the bridge to illustrate its utility. I separate results that follow from the reduction alone (Section 5.1) from those that additionally import prior Stack-Theoretic machinery (Section 5.2).

## 5.1 Direct corollaries of the reduction

The following results use only Theorem 11 and basic partition combinatorics. No Stack-Theoretic imports are required.

### 5.1.1 A regret-based partial order on abstractions

The bridge induces a natural ordering on representations.

**Proposition 24** (Regret ordering). *Define $M_1 \preceq_\rho M_2$ iff $K_\rho(M_1) \geq K_\rho(M_2)$. This partial order is consistent with the partition refinement lattice: if $\mathcal{C}_{M_2}|_{I_+}$ refines $\mathcal{C}_{M_1}|_{I_+}$, then $K_\rho(M_2) \leq K_\rho(M_1)$. The inequality is strict whenever the refinement splits at least one mixed cell on $I_+$ into subcells in which the minority class flips (i.e. the label achieving $\min\{A, B\}$ differs between subcells).*

*Proof.* If $\mathcal{C}_{M_2}$ refines $\mathcal{C}_{M_1}$, then every cell $C_1 \in \mathcal{C}_{M_1}$ is partitioned into subcells $C_2^{(j)} \in \mathcal{C}_{M_2}$. By the cellwise formula,

$$\min\{A_{C_1}, B_{C_1}\} \geq \sum_j \min\{A_{C_2^{(j)}}, B_{C_2^{(j)}}\},$$

since $\min\{a + a', b + b'\} \geq \min\{a, b\} + \min\{a', b'\}$ for non-negative reals. Summing gives $K_\rho(M_1) \geq K_\rho(M_2)$. For strictness, suppose a mixed cell $C_1$ with $A_{C_1} > B_{C_1}$ splits into subcells $C_2, C_2'$ where $A_{C_2} > B_{C_2}$ but $A_{C_2'} < B_{C_2'}$. Then

$$\min\{A_{C_1}, B_{C_1}\} = B_{C_2} + B_{C_2'} > B_{C_2} + A_{C_2'} = \min\{A_{C_2}, B_{C_2}\} + \min\{A_{C_2'}, B_{C_2'}\},$$

so the contribution from this cell is strictly smaller after refinement. $\square$

**Intuition.** This gives the state-abstraction and bisimulation communities a new tool: a regret-based partial order on abstractions that complements the existing bisimulation-metric order of Ferns et al. (2004). Finer representations have lower or equal $K_\rho$, and the ordering indicates when refinement helps. The ordering also complements the Richens-Everitt qualitative characterisation (Richens & Everitt, 2024). Their result says robust agents must learn causal models. The regret ordering says how much it costs if the learned representation falls short of the causal partition.

### 5.1.2 Representation convergence in RL language

The regret ordering has a clean endpoint. At zero $K_\rho$, every cell is pure, and the coarsest pure partition is unique. This gives a representation convergence theorem stated entirely in RL language.

**Theorem 25** (Representation convergence). *Fix a finite evaluation support $(\mathcal{X}, \mu, y, \rho)$ and let $I_+ = \{i : \mu_i \rho_i > 0\}$ be the informative support. Let $\mathcal{T}_+ := \{T_i : i \in I_+\}$. Define $\mathcal{P}_+^\star := \{\{i \in I_+ : T_i = t, y_i = b\} : t \in \mathcal{T}_+, b \in \{0, 1\}\} \setminus \{\varnothing\}$, the coarsest partition of $I_+$ that separates test values and optimal labels.*

1. *$K_\rho(M) = 0$ if and only if $\mathcal{C}_M|_{I_+}$ refines $\mathcal{P}_+^\star$.*

2. *If $M$ is minimal under partition coarsening on $I_+$ among the zero-regret representations, then $\mathcal{C}_M|_{I_+} = \mathcal{P}_+^\star$.*

3. *Any two minimal zero-regret representations induce the same partition on $I_+$ and differ there only by a relabelling of representation states.*

*Proof.* By Theorem 11, $K_\rho(M) = 0$ iff $\min\{A_C, B_C\} = 0$ for every cell $C$. On $I_+$ this means each cell contains only one optimal label at its fixed test value, which is $\mathcal{C}_M|_{I_+} \preceq \mathcal{P}_+^\star$. Conversely, points outside $I_+$ carry zero deletion weight, so refinement there does not affect $K_\rho$. For part 2, $\mathcal{P}_+^\star$ is itself zero-cost because each of its cells is pure. If a minimal zero-cost $M$ were strictly finer than $\mathcal{P}_+^\star$ on $I_+$, coarsening to $\mathcal{P}_+^\star$ would preserve zero cost, contradicting minimality. Part 3 follows because equality of quotients on $I_+$ means there is a bijection between nonempty representation states, which is a relabelling. $\square$

**Intuition.** Any two minimal zero-regret representations must carve the informative support the same way. The proof uses only the cellwise decomposition from Theorem 11 and basic partition logic.

### 5.1.3 Rate-distortion interpretation

$K_\rho(M)$ has a natural information-theoretic reading. The representation $M$ is a lossy compression of histories. The margin-weighted forgetting cost $K_\rho(M)$ is the residual relevant information that the bottleneck has destroyed, weighted by diagnostic importance. In the language of the information bottleneck (Tishby et al., 2000), $K_\rho(M)$ is the Bayes error of the sufficient statistic induced by $M$, expressed as a margin-weighted deletion cost. The bridge theorem says this is the regret cost of that compression. Representations with lower $K_\rho$ are better compressions in the sense that matters for decision-making: they preserve the distinctions that actually affect regret and discard the ones that do not.

### 5.1.4 Stability under margin estimation error

In practice the conditional probabilities $p_i$ are estimated from data, so the margin weights $\rho_i$ are known only approximately. The following shows that $K_\rho$ is Lipschitz in the margin estimates, so small estimation errors produce small diagnostic errors.

**Proposition 26** (Lipschitz stability of $K_\rho$)**.** *Let $\hat\rho_1, \ldots, \hat\rho_n$ be perturbed margin weights. Define $\hat K_\rho(M)$ by replacing $\rho_i$ with $\hat\rho_i$ in Theorem 9. Then*

$$\left| K_{\hat\rho}(M) - K_\rho(M) \right| \leq \sum_{i=1}^n \mu_i \left| \hat\rho_i - \rho_i \right|.$$

*Moreover, since $\rho_i$ is a function of $p_i$ with $|d\rho_i/dp_i| \leq 4$ everywhere on $(0,1)$, estimation error in the conditional probabilities propagates as*

$$\left| K_{\hat\rho}(M) - K_\rho(M) \right| \leq 4 \sum_{i=1}^n \mu_i \left| \hat p_i - p_i \right|.$$

*Proof.* For each cell $C$, define $\hat A_C := \sum_{i \in C,\, y_i=1} \mu_i \hat\rho_i$ and $\hat B_C$ analogously. Since $|\min\{a', b'\} - \min\{a, b\}| \leq |a' - a| + |b' - b|$ for non-negative reals,

$$\left| \min\{\hat A_C, \hat B_C\} - \min\{A_C, B_C\} \right| \leq |\hat A_C - A_C| + |\hat B_C - B_C| \leq \sum_{i \in C} \mu_i |\hat\rho_i - \rho_i|.$$

Summing over cells gives the first bound. For the second, note that for $p > \frac{1}{2}$ we have $\rho = (2p-1)/p$, so $d\rho/dp = 1/p^2 \leq 4$; for $p < \frac{1}{2}$ we have $\rho = (1-2p)/(1-p)$, so $|d\rho/dp| = 1/(1-p)^2 \leq 4$. Hence $|\hat\rho_i - \rho_i| \leq 4|\hat p_i - p_i|$. $\qquad\square$

**Intuition.** The diagnostic $K_\rho(M)$ degrades gracefully under estimation noise. With $N$ samples per support point, standard concentration gives $|\hat p_i - p_i| = O(N^{-1/2})$, so $|\hat K_\rho - K_\rho| = O(N^{-1/2})$ as well.

## 5.2 Cross-framework consequences

The following results combine Theorem 11 with prior Stack-Theoretic machinery. Each one is flagged with the additional assumption it requires.

### 5.2.1 Free-energy floor from irreducible regret

Combining Theorem 22 with the Law of the Stack (Bennett, 2024b) requires one extra assumption, because the bridge uses weighted retained mass rather than ordinary extension cardinality. In the unweighted Law of the Stack, under a uniform viability prior at layer $i + 1$, the base-2 free-energy proxy satisfies

$F_2 \geq \log_2 |\text{Ext}(\mu)| - |\text{Ext}(\pi^i)|$. The corollary below uses the corresponding weighted lifted version, where counting measure on the diagnostic layer is replaced by

$$\nu(A) := \sum_{u_i \in A \cap U} w_i.$$

**Corollary 27** (Free-energy floor from regret). *In the setting of Theorem 22, suppose the lifted diagnostic task is embedded as a layer in a Stack-Theoretic hierarchy. Assume the Law of the Stack is applied in the weighted lifted language with counting measure replaced by the finite measure $\nu$ above. Equivalently, assume the layer-above free-energy proxy satisfies*

$$F_{2,\nu} \geq C_\nu - \nu(\text{Ext}(\pi) \cap U)$$

*for the relevant parent-capacity term $C_\nu$. Then the best admissible policy under representation $M$ satisfies*

$$F_{2,\nu} \geq C_\nu - W^\star(M) = C_\nu - (Z - K_\rho(M)).$$

*In particular, relative to the same weighted lifted layer and the same constant $C_\nu$, any positive irreducible regret $K_\rho(M) > 0$ raises the free-energy floor by $K_\rho(M)$ compared with the zero-regret case.*

*Proof.* By the weighted lifted Law-of-the-Stack assumption, $F_{2,\nu} \geq C_\nu - \nu(\text{Ext}(\pi) \cap U)$ at the diagnostic layer. The maximum weighted extension mass of any admissible policy on the lifted task is $W^\star(M)$. By Theorem 22, $W^\star(M) = Z - K_\rho(M)$. Substituting gives the display. $\square$

**Intuition.** This links RL regret to variational free energy once the Stack-Theoretic layer is interpreted with the same weighted measure used by the regret bridge. The extra assumption is needed because $W^\star(M)$ is a weighted retained mass, not an ordinary cardinality. A bad representation at a lower layer puts a floor under free energy at the layer above.

### 5.2.2 Generalisation probability from regret

The free-energy corollary imports Stack-Theoretic structure but stays in the language of bounds. The following corollary converts the regret floor into a generalisation probability, giving $K_\rho$ a direct predictive semantics that the RL identity alone does not provide.

**Corollary 28** (Generalisation probability from regret). *In the setting of Theorem 22, choose independent relevance probabilities $r_i = 1 - e^{-\mu_i \rho_i}$ as in Theorem 22. Under this prior, the probability that the optimal $M$-based policy generalises to a new set of diagnostic demands is*

$$\Pr(\text{optimal policy generalises}) = e^{-K_\rho(M)}.$$

*Proof.* By Theorem 21, generalisation occurs when no deleted unseen output becomes relevant. The optimal admissible policy deletes the cheaper label class in each cell, removing total weight $K_\rho(M)$ from the unseen region. Under the independent prior with $r_i = 1 - e^{-w_i}$, the generalisation probability of any admissible policy $\vartheta$ is

$$\prod_{u_i \notin \text{Ext}(\vartheta) \cap U} (1 - r_i) = \prod_{u_i \notin \text{Ext}(\vartheta) \cap U} e^{-w_i} = e^{-\sum_{u_i \notin \text{Ext}(\vartheta) \cap U} w_i}.$$

For the optimal policy, the exponent is $K_\rho(M)$. $\square$

**Intuition.** This is the payoff of the bridge as a bidirectional connection, not merely a dictionary. The RL side provides $K_\rho(M)$ as a regret floor. The Stack-Theoretic side, via the independent-prior generalisation model, converts that regret floor into a generalisation probability. Neither framework alone gives both quantities. A representation with $K_\rho = 0.02$ generalises with probability $e^{-0.02} \approx 0.98$ under the canonical prior. One with $K_\rho = 0.20$ generalises with probability $e^{-0.20} \approx 0.82$. The exponential sensitivity to $K_\rho$ makes the diagnostic practically meaningful: small differences in irreducible regret translate into measurable differences in generalisation reliability.

### 5.2.3 Multi-agent coordination cost

Combining Theorem 11 with the contravariance result (Appendix B), we quantify the cost of sharing a representation across subsystems with incompatible diagnostic needs.

**Proposition 29** (Coordination cost). *Let $N$ subsystems each face a diagnostic family on the same support with the same single test $T$. Let $\mathcal{P}_j^\star$ be the coarsest pure partition for subsystem $j$, with $|\mathcal{P}_j^\star| = k_j$ cells. A shared representation that achieves zero regret for all $N$ subsystems simultaneously requires at least $|\mathcal{P}_1^\star \vee \cdots \vee \mathcal{P}_N^\star|$ cells, where $\vee$ denotes the common refinement. In the worst case this is $\prod_{j=1}^N k_j$.*

*Proof.* By Theorem 11, zero regret for subsystem $j$ requires the shared partition to refine $\mathcal{P}_j^\star$. The coarsest partition refining all $N$ is the common refinement $\mathcal{P}_1^\star \vee \cdots \vee \mathcal{P}_N^\star$. Its size is bounded above by $\prod_j k_j$ because each cell is determined by its class in each subsystem's partition. The bound is achieved when the partitions are independent (no two agree on the grouping of any pair of support points). $\square$

**Intuition.** Subsystems with incompatible diagnostic needs impose a combinatorial tax on shared representations. This connects to federated learning, where client drift (Karimireddy et al., 2020) has the same structure, and to the biological literature on cancer-like coordination failure (Davies & Lineweaver, 2011; Levin, 2019; Fields & Levin, 2020; Bennett, 2024b).

**Example 30** (Coordination cost in a two-subsystem scenario). Consider a domain with 6 support points under one common test $T$. Subsystem $A$ requires the partition $\mathcal{P}_A^\star = \{\{1,2\}, \{3,4,5,6\}\}$ ($k_A = 2$), while subsystem $B$ requires $\mathcal{P}_B^\star = \{\{1,2,3\}, \{4,5,6\}\}$ ($k_B = 2$). Each subsystem needs only 2 representation states, but a shared representation that achieves zero regret for both must have at least $|\mathcal{P}_A^\star \vee \mathcal{P}_B^\star| = |\{\{1,2\}, \{3\}, \{4,5,6\}\}| = 3$ states. In the worst case with $N$ such subsystems, the shared representation may require up to $\prod_j k_j$ states, while any individual subsystem needs only $k_j$. This gap is the coordination tax.

## 6 Diagnostic experiments

The theorems above are exact algebraic identities, so they do not require empirical validation. The purpose of this section is to show that $K_\rho(M)$ is informative in practice: that the bridge identity holds under native Stack-Theoretic computation, that $K_\rho$ discriminates between representations where simpler diagnostics do not, and that the aliasing–execution decomposition tracks representation quality during training.

### 6.1 Two-sided verification on Boolean domain

Here the diagnostic task is encoded natively using the "Stack Theory Suite" code library (Bennett, 2025d). The environment has $n_{\text{bits}} = 5$ Boolean variables, giving $2^5 = 32$ states. The vocabulary contains 10 programs (bit-tests: $b_i{=}0$ and $b_i{=}1$ for each bit $i$), giving an induced language $|L_\mathfrak{v}| = 243$. For each trial, a random structured Boolean full-state label $Y : \{0,1\}^5 \to \{0,1\}$ is generated that depends on 2–3 randomly chosen bits. Representations are defined by which bits the agent observes (0 through 5). The diagnostic support points are partial observation histories rather than necessarily full Boolean states. For a partial observation pattern $s$, the diagnostic probability $p(s)$ is the posterior probability that $Y = 1$ after marginalising over full Boolean states consistent with $s$. Thus full-state labels are deterministic, while diagnostic probabilities can still be near $1/2$.

The bridge is verified two-sided through independent code paths. *Side A (RL):* cells are computed by direct bit inspection; $K_\rho$ is computed via the cellwise formula $\sum_C \min\{A_C, B_C\}$. *Side B (Stack Theory):* for each possible observation pattern, the corresponding STS statement is built by conjoining the appropriate bit-test programs. Its truth set is computed via `Statement.truth_set()`. The states in the truth set define the cell membership. $K_\rho$ is then recomputed from the STS-derived cells. Both the cell structures and the resulting $K_\rho$ values are compared. Over 100 trials $\times$ 6 representation levels = 600 checks, cell membership agrees in every case, and $K_\rho$ matches to numerical precision. The two-sided agreement confirms that the Stack-Theoretic reformulation is not a post-hoc relabelling: the STS truth-set computation and the RL cellwise computation produce identical cell structures through entirely independent formal machinery.

Table 3: Boolean classification experiment (5 bits, 32 states, 100 trials). $K_\rho$ is computed via RL cellwise arithmetic (Side A) and independently via STS truth-set enumeration (Side B); both agree to numerical precision in all 600 checks. Cell counts are deterministic given the number of observed bits.

| Bits observed | $K_\rho(M)$ | Raw impurity | Accuracy | Cells |
|---:|---|---|---|---:|
| 0 | $0.102 \pm 0.008$ | $0.318 \pm 0.015$ | $0.68 \pm 0.01$ | 1 |
| 1 | $0.079 \pm 0.007$ | $0.285 \pm 0.014$ | $0.71 \pm 0.01$ | 2 |
| 2 | $0.047 \pm 0.007$ | $0.247 \pm 0.015$ | $0.75 \pm 0.02$ | 4 |
| 3 | $0.019 \pm 0.005$ | $0.191 \pm 0.015$ | $0.81 \pm 0.01$ | 8 |
| 4 | $0.000 \pm 0.000$ | $0.124 \pm 0.016$ | $0.88 \pm 0.02$ | 16 |
| 5 | $0.000 \pm 0.000$ | $0.000 \pm 0.000$ | $1.00 \pm 0.00$ | 32 |

Table 4: Discretised MLP encoder on a 16-state POMDP (20 trials). Finer discretisation reduces both $K_\rho$ and raw impurity, but $K_\rho$ drops faster: at 6 bins, $K_\rho$ is a factor of 5.5× smaller than raw impurity.

| Bins | $K_\rho(M)$ | Raw impurity | Accuracy | Cells |
|---:|---|---|---|---:|
| 2 | $0.0097 \pm 0.0004$ | $0.041 \pm 0.001$ | $0.959 \pm 0.001$ | 166 |
| 3 | $0.0101 \pm 0.0008$ | $0.038 \pm 0.002$ | $0.962 \pm 0.002$ | 172 |
| 4 | $0.0057 \pm 0.0004$ | $0.026 \pm 0.001$ | $0.974 \pm 0.001$ | 343 |
| 5 | $0.0036 \pm 0.0003$ | $0.017 \pm 0.001$ | $0.983 \pm 0.001$ | 378 |
| 6 | $0.0022 \pm 0.0002$ | $0.012 \pm 0.001$ | $0.989 \pm 0.001$ | 517 |
| 8 | $0.0009 \pm 0.0002$ | $0.005 \pm 0.001$ | $0.995 \pm 0.001$ | 630 |
| 10 | $0.0003 \pm 0.0000$ | $0.002 \pm 0.000$ | $0.998 \pm 0.000$ | 700 |
| 15 | $0.0001 \pm 0.0000$ | $0.000 \pm 0.000$ | $1.000 \pm 0.000$ | 763 |

Table 3 shows that $K_\rho$ and raw impurity behave differently. $K_\rho$ reaches zero at 4 observed bits: the remaining unobserved bit creates cells that are impure but whose impurity carries zero margin weight (the minority class within each cell has $\rho_i \approx 0$ because the posterior is close to a coin flip on that bit). Raw impurity remains positive until all 5 bits are observed. In a separate comparison of 200 pairs of 2-bit versus 3-bit representations, $K_\rho$ discriminated between representations that had similar accuracy ($< 5\%$ difference) in 21% of cases.

## 6.2 Discretised encoder at varying granularity

To demonstrate the diagnostic value of $K_\rho$ at moderate scale, we construct a structured POMDP with $|\mathcal{S}| = 16$ hidden states, $|\mathcal{O}| = 6$ observations, and observation sequences of length $L = 8$. The first 8 states have label $y = 1$; the remaining 8 have label $y = 0$. Observation distributions are block-structured so that states in the same class share similar emission profiles, with cross-block transitions occurring infrequently.

I train a two-layer MLP (hidden dimension 4, tanh activation) to convergence on each of 20 random POMDPs, then evaluate the learned representation at 8 levels of discretisation granularity (from 2 to 15 bins per hidden dimension). Finer discretisation yields more cells and a more refined partition of the observation-history space.

Table 4 shows that $K_\rho$ is consistently smaller than raw impurity across all granularities, because it discounts aliasing on near-tie diagnostic tests. At 6 bins per dimension, $K_\rho = 0.002$ while raw impurity is still 0.012, a 5.5× ratio. Both quantities eventually approach zero at fine granularity, but the gap at intermediate levels is where the diagnostic is most useful: a representation that looks impure by raw vote may already have near-zero aliasing cost on the tests that actually matter for decision-making.

## 6.3 Architecture comparison

To test whether $K_\rho$ provides actionable information beyond accuracy, we compare four representation architectures on the same 16-state structured POMDP: a GRU (hidden dimension 4, trained for 300 epochs

Table 5: Architecture comparison on a 16-state POMDP (20 trials). The GRU has the lowest $K_\rho$ but the MLP has the highest accuracy. In 35% of trials, the architecture with the best $K_\rho$ differs from the one with the best accuracy.

| Architecture | $K_\rho(M)$ | Accuracy | Raw impurity | Cells |
|---|---|---|---|---|
| GRU | $0.0031 \pm 0.0004$ | $0.978 \pm 0.002$ | $0.022 \pm 0.002$ | 50 |
| MLP | $0.0039 \pm 0.0004$ | $0.982 \pm 0.001$ | $0.018 \pm 0.001$ | 371 |
| Last-obs | $0.0683 \pm 0.0026$ | $0.838 \pm 0.004$ | $0.162 \pm 0.004$ | 6 |
| Random | $0.1679 \pm 0.0057$ | $0.755 \pm 0.006$ | $0.245 \pm 0.006$ | 212 |

Table 6: Decomposition for a learned MLP during training on the 16-state POMDP ($L = 8$, hidden dim 4, 20 trials). The ratio $K_\rho/(K_\rho + \mathrm{exec.\,cost})$ drops from 0.30 to 0.03 in the first 100 epochs, making the transition from representational bottleneck to policy bottleneck visible.

| Epoch | $K_\rho(M_\theta)$ | Execution cost | $K_\rho$ / total | Accuracy |
|---|---|---|---|---|
| 0 | $0.112 \pm 0.005$ | $0.268 \pm 0.005$ | 0.30 | $0.49 \pm 0.01$ |
| 50 | $0.012 \pm 0.001$ | $0.158 \pm 0.005$ | 0.07 | $0.81 \pm 0.00$ |
| 100 | $0.003 \pm 0.000$ | $0.097 \pm 0.002$ | 0.03 | $0.83 \pm 0.00$ |
| 150 | $0.003 \pm 0.000$ | $0.090 \pm 0.001$ | 0.03 | $0.83 \pm 0.00$ |
| 250 | $0.003 \pm 0.000$ | $0.086 \pm 0.001$ | 0.03 | $0.83 \pm 0.00$ |
| 500 | $0.003 \pm 0.000$ | $0.085 \pm 0.001$ | 0.04 | $0.83 \pm 0.00$ |

on sequences), an MLP (hidden dimension 4, trained for 400 epochs on flattened one-hot inputs), a last-observation-only baseline, and a random projection (tanh of a fixed random linear map). All representations are discretised with 5 bins per hidden dimension.

Table 5 reveals that the GRU achieves the lowest irreducible regret ($K_\rho = 0.003$), while the MLP achieves the highest accuracy (0.982). The MLP achieves higher end-to-end accuracy even though its discretised representation has slightly higher irreducible aliasing cost. This shows that accuracy includes effects of the trained classifier and the continuous hidden geometry available to that classifier, whereas $K_\rho$ isolates the aliasing cost of the chosen representation-test quotient after discretisation. In 35% of trials, the architecture with the best $K_\rho$ differs from the one with the best accuracy. This confirms that $K_\rho$ captures a distinct aspect of representation quality, namely the structural aliasing cost, that accuracy conflates with policy optimisation.

### 6.4 Learned representation during training

To demonstrate the decomposition as a training diagnostic, we train a two-layer MLP classifier (hidden dimension 4, tanh activation, discretised to 5 bins per hidden unit) on the 16-state structured POMDP with $|\mathcal{O}| = 6$ observations and observation sequences of length $L = 8$. The discretised hidden-layer activations define the representation-test cells.

Table 6 reports $K_\rho(M_\theta)$, execution cost, and their ratio at selected training epochs, averaged over 20 trials.

$K_\rho$ drops by a factor of $37\times$ in the first 100 epochs (from 0.112 to 0.003) as the hidden layer learns to separate the two state classes. After that, $K_\rho$ plateaus near zero and further improvements come from execution cost reduction. The ratio $K_\rho/(K_\rho + \mathrm{exec.\,cost})$ falls from 0.30 to 0.03 during this period, making the bottleneck transition quantitatively visible. Early in training, the representation is the bottleneck. Later, the policy is the bottleneck. The decomposition makes that transition readable from a single diagnostic, which is the practical contribution of the bridge for representation evaluation.

# 7 Towards representation learning via the bridge

The bridge theorem gives exact tools for computing and decomposing the representational component of regret. A natural question is whether these tools can guide representation learning during RL training. This section discusses the prospects and obstacles.

## 7.1 Auxiliary losses derived from the bridge

The identity suggests a family of auxiliary losses for training recurrent encoders on partially observable tasks. Given a hidden-state predictor layered on top of the encoder, one can weight the prediction loss by the margin weight $\rho_i$, focusing the encoder on states where the diagnostic distinction is decision-relevant. Alternatively, one can apply an MDL compression penalty (KL divergence toward a maximum-entropy prior) to suppress noise in spare representational capacity. These strategies operate on the *representation*, not on the policy's extension, and are therefore representation-level surrogates for the formal weakness-maximisation principle rather than implementations of it.

Preliminary experiments on four small partially observable environments (a binary-context corridor, T-Maze, NoisyCartPole with hidden velocities, and a symbol-recall task) suggest that the optimal strategy is task-dependent: margin weighting helps when the diagnostic and control objectives are aligned (as in T-Maze, where the hidden label *is* the decision-relevant distinction), while MDL compression helps when the encoder has excess capacity relative to the task's information content. No single strategy dominates. This is consistent with the expectation that representation-level surrogates cannot capture the policy-level quantity that the weakness theorems optimise. Full experimental code and results are included in the supplementary material.

## 7.2 The open problem: operationalising weakness

The weakness-maximisation theorems (Bennett, 2025c;d) prove that the weakest correct policy, namely the one that maximises $|\text{Ext}(\pi)|$ in an embodied language $L_{\mathfrak{v}}$, is optimal for generalisation under the maximally uninformative prior. A neural network policy is a total function from observations to action distributions. It specifies an output for every input. There are no "unset bits" whose completions could be counted, so the extension is trivial and weakness is undefined.

The central difficulty is that weakness is defined relative to a language $L_{\mathfrak{v}}$ that determines which completions count. A neural network does not have a native analogue of this structure. One possible approach is a hybrid architecture in which a neural encoder discretises observations into a finite vocabulary, and exact weakness computation is then performed in the discrete domain using existing Stack-Theoretic machinery. Developing and evaluating such an architecture, and scaling the auxiliary-loss strategies to standard partially observable benchmarks (Morad et al., 2023), are the natural next steps.

# 8 Conclusion

At fixed representation-test quotient, minimum average normalised regret is margin-weighted forgetting. For an actual policy, total regret decomposes into irreducible aliasing cost plus avoidable within-cell misreporting. In Stack-Theoretic form, the same quantity is a weighted weakness deficit on a lifted diagnostic task.

The identity yields a representation-convergence theorem in pure RL language, a regret-based partial order on abstractions, Lipschitz stability under margin estimation error, and connections to free energy and multi-agent coordination, along with the multi-class generalisation to $K > 2$ outcomes. A cross-framework corollary converts the regret floor into a generalisation probability ($e^{-K_\rho(M)}$ under the canonical independent prior), giving the bridge bidirectional utility: the RL side supplies the regret floor, the Stack-Theoretic side converts it into a generalisation guarantee. The decomposition also serves as a practical training diagnostic: on controlled POMDPs, $K_\rho$ isolates representational quality where accuracy does not, and the aliasing–execution ratio makes the transition from representational bottleneck to policy bottleneck quantitatively visible.

The weakness-maximisation theorems (Bennett, 2025c;d) predict that the least-committal correct policy maximises generalisation probability. However, the formal object they optimise, namely the extension of

a policy in an embodied language, does not have a direct analogue in neural network function approximation. Preliminary experiments with bridge-derived auxiliary losses (margin-weighted state prediction, MDL compression) suggest that the optimal representation-level strategy is task-dependent and that no single surrogate dominates. Operationalising formal weakness in continuous function approximation, and developing the hybrid discrete–neural architecture needed to do so, remain the central open problems.

## Broader impact statement

This paper is primarily theoretical, with controlled POMDP experiments on small structured environments. It does not introduce a deployed capability. Its primary positive impact is conceptual and methodological: the decomposition serves as a diagnostic tool for evaluating learned representations, and the derived results connect RL, active inference, and biological coordination through one shared defect variable. Bridge papers can overstate equivalence, novelty, or priority. I mitigate that risk by keeping the formal claims at the level of the representation-test quotient, stating explicitly what earlier work did and did not prove, noting where corollaries follow by direct substitution, and distinguishing representation-level strategies from the formal weakness-maximisation principle.

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

## A  Measurable reduction

The finite theorem has a direct measurable analogue. Let $(\mathcal{X}, \Sigma, \mu)$ be a probability space of history-test points $x = (h, T)$. Let

$$\zeta := (M(h), T)$$

be the observable representation-test variable. Assume $y : \mathcal{X} \to \{0, 1\}$ and $\rho : \mathcal{X} \to [0, 1]$ are measurable. Define

$$a(z) := \mathbb{E}\big[\rho \, \mathbf{1}\{y = 1\} \mid \zeta = z\big], \qquad b(z) := \mathbb{E}\big[\rho \, \mathbf{1}\{y = 0\} \mid \zeta = z\big].$$

Any $M$-based policy is determined by a measurable map $q : \mathcal{Z} \to [0, 1]$. Its expected normalised regret is

$$R(q) := \mathbb{E}\Big[\rho\big((1 - q(\zeta))\mathbf{1}\{y = 1\} + q(\zeta)\mathbf{1}\{y = 0\}\big)\Big].$$

**Proposition 31** (Measurable reduction). *With the notation above,*

$$\inf_{q:\mathcal{Z} \to [0,1] \ measurable} R(q) = \mathbb{E}\big[\min\{a(\zeta), b(\zeta)\}\big].$$

*An optimal measurable selector is given by*

$$q^\star(z) = \begin{cases} 1 & a(z) > b(z), \\ 0 & a(z) < b(z), \\ any \ value \ in \ [0, 1] & a(z) = b(z). \end{cases}$$

*Proof.* By the tower property,

$$R(q) = \mathbb{E}\big[a(\zeta)(1 - q(\zeta)) + b(\zeta)q(\zeta)\big].$$

For each fixed $z$, the integrand is affine in $q(z)$. Its pointwise minimum over $q(z) \in [0, 1]$ is therefore $\min\{a(z), b(z)\}$. Choosing $q^\star$ pointwise attains that minimum almost surely. Integrability is immediate because $0 \le a, b \le \mathbb{E}[\rho \mid \zeta] \le 1$. $\qquad\square$

This is the conditional-expectation form of the finite theorem. In the finite support case, $\zeta$ takes one value per representation-test cell, $a(C) = A_C/\mu(C)$ and $b(C) = B_C/\mu(C)$, so the outer expectation supplies the factor $\mu(C)$ and Theorem 31 reduces to $\sum_C \min\{A_C, B_C\}$. In measurable language, $\mathbb{E}[\min\{a(\zeta), b(\zeta)\}]$ is the weighted impurity that remains after quotienting by representation-test cell. It is the natural measurable version of $K_\rho(M)$.

# B   Contravariance and splintering on the diagnostic quotient

The bridge theorem already contains a small contravariance result. Let

$$I_+ := \{i : \mu_i \rho_i > 0\}, \qquad \mathcal{T}_+ := \{T_i : i \in I_+\}.$$

Define the *coarsest pure diagnostic partition on the informative support*

$$\mathcal{P}_+^\star := \big\{ \{i \in I_+ : T_i = t, \ y_i = b\} : t \in \mathcal{T}_+, \ b \in \{0,1\} \big\} \setminus \{\varnothing\}.$$

So $\mathcal{P}_+^\star$ groups positive-weight support points only by test and optimal label. Write $\mathcal{C} \preceq \mathcal{D}$ when every cell of partition $\mathcal{C}$ is contained in some cell of $\mathcal{D}$.

**Proposition 32** (Minimal zero-cost quotients are unique up to recoding)**.** *For a fixed support and label map $y$, the following hold.*

1. *$K_\rho(M) = 0$ if and only if the restriction of $\mathcal{C}_M$ to $I_+$ satisfies $\mathcal{C}_M|_{I_+} \preceq \mathcal{P}_+^\star$.*

2. *If $M$ is minimal among the zero-cost representations under partition coarsening on $I_+$, then $\mathcal{C}_M|_{I_+} = \mathcal{P}_+^\star$.*

3. *Consequently, any two minimal zero-cost representations induce the same quotient on the informative support and differ there only by a recoding of representation states.*

*Proof.* If $K_\rho(M) = 0$, then every cell $C \in \mathcal{C}_M$ must satisfy $\min\{A_C, B_C\} = 0$ by Theorem 11. So on the positive-weight support $I_+$ each cell contains only one optimal label at its fixed test value, which is the statement that $\mathcal{C}_M|_{I_+} \preceq \mathcal{P}_+^\star$. The converse is immediate from the definition of $K_\rho(M)$ because points outside $I_+$ carry zero deletion weight.

For part 2, note that $\mathcal{P}_+^\star$ itself is zero-cost because every one of its cells is pure by construction. If a zero-cost representation $M$ were minimal but strictly finer than $\mathcal{P}_+^\star$ on $I_+$, then coarsening $\mathcal{C}_M|_{I_+}$ to $\mathcal{P}_+^\star$ would preserve zero cost, contradicting minimality. So $\mathcal{C}_M|_{I_+} = \mathcal{P}_+^\star$. Part 3 follows because equality of quotients on the informative support means there is a bijection between the nonempty representation states induced by the two minimal representations there. That bijection is a recoding. □

**Intuition.**   Shared task pressure drives away weighted impurity. At zero impurity there is one coarsest way to preserve the required distinctions. Minimal competent agents therefore converge on the same informative quotient, even if they use different internal names for its cells.

**Example 33** (Shared pressure versus splintering)**.** Take three support points $x_1, x_2, x_3$ under one common test $T$. Subsystem $A$ has optimal labels

$$y^A(x_1) = 1, \qquad y^A(x_2) = 0, \qquad y^A(x_3) = 0,$$

so its coarsest pure quotient is

$$\mathcal{P}_{A,+}^\star = \{\{x_1\}, \{x_2, x_3\}\}.$$

Subsystem $B$ has labels

$$y^B(x_1) = 0, \qquad y^B(x_2) = 1, \qquad y^B(x_3) = 0,$$

so its coarsest pure quotient is

$$\mathcal{P}_{B,+}^\star = \{\{x_2\}, \{x_1, x_3\}\}.$$

These minimal zero-cost quotients do not coincide. If each subsystem separately minimises its own weighted forgetting cost, they converge to different recoding classes. If they are forced to share one common zero-cost quotient, they must instead move to the finer common refinement

$$\{\{x_1\}, \{x_2\}, \{x_3\}\}.$$

This is the abstract pattern behind the contrast used in the main text. Shared task pressure yields contravariant convergence to one common quotient. Split task pressure yields divergent locally sufficient quotients, which is the toy formal template for splintering and cancer-like coordination failure.

# C  Imported Stack-Theoretic machinery

This appendix gives self-contained formal restatements of the Stack-Theoretic results that the main text imports. Each restatement uses the same notation and hypotheses as the master appendix (Bennett, 2025d). Proofs are sketched in plain English. Full proofs are in the cited sources.

This appendix is intended to make the cross-framework corollaries in Section 5.2 verifiable without having to reconstruct definitions from elsewhere in the corpus. A reader who has trusted the imports already can skip to Section C.6 for the genealogy summary.

## C.1  Minimal preliminaries

The following definitions are sufficient to state every imported result. They are taken verbatim from Bennett (2024b) and Bennett (2025d).

**Definition 34** (Environment, vocabulary, embodied language). An *environment* is a nonempty set $\Phi$ of mutually exclusive states. A *program* is a subset $p \subseteq \Phi$. A *vocabulary* is a finite set $\mathfrak{v} \subseteq 2^\Phi$ of programs. The *embodied language* induced by $\mathfrak{v}$ is the satisfiable fragment

$$L_\mathfrak{v} := \big\{\, l \subseteq \mathfrak{v} \ \big|\ \bigcap_{p \in l} p \neq \varnothing \,\big\}.$$

Elements of $L_\mathfrak{v}$ are called *statements*. A statement is true at state $\phi$ when $\phi$ lies in the intersection of its programs.

Note that $L_\mathfrak{v}$ is the satisfiable fragment of $2^\mathfrak{v}$, not all of $2^\mathfrak{v}$. Statements correspond to physical configurations a body can be in, and most configurations are mutually exclusive. This restriction is what stops weakness from collapsing into a pure length proxy.

**Definition 35** (Extension and weakness). For $l \in L_\mathfrak{v}$, the *extension* of $l$ is

$$\mathrm{Ext}(l) := \{\, y \in L_\mathfrak{v} \mid l \subseteq y \,\}.$$

The *weakness* of $l$ is $w(l) := |\mathrm{Ext}(l)|$, the number of completions of $l$ within $L_\mathfrak{v}$. A statement $l_1$ is *weaker* than $l_2$ when $w(l_1) > w(l_2)$.

Weakness is intrinsic relative to a fixed embodied language $L_\mathfrak{v}$. It is not a function of external description length. The numerical weakness score $|\mathrm{Ext}(l)|$ depends on the chosen embodied vocabulary. The encoding-invariant claim concerns the optimality of the weakness rule inside each finite vocabulary, not equality of weakness scores across vocabularies.

**Definition 36** ($\mathfrak{v}$-task, correct policy). For $X \subseteq L_\mathfrak{v}$, write

$$\mathrm{Ext}(X) := \bigcup_{x \in X} \mathrm{Ext}(x).$$

A $\mathfrak{v}$-*task* is a pair $\alpha = \langle I_\alpha, O_\alpha \rangle$ of finite subsets, with $I_\alpha \subseteq L_\mathfrak{v}$ the inputs and $O_\alpha \subseteq \mathrm{Ext}(I_\alpha)$ the correct outputs. A *policy* is any statement $\pi \in L_\mathfrak{v}$. The policy is *correct* for $\alpha$ iff

$$\mathrm{Ext}(I_\alpha) \cap \mathrm{Ext}(\pi) = O_\alpha.$$

Write $\Pi_\alpha$ for the set of correct policies for $\alpha$. Tasks $\alpha, \omega$ satisfy $\alpha \sqsubset \omega$ when $I_\alpha \subseteq I_\omega$ and $O_\alpha \subseteq O_\omega$ (the parent extends the child).

## C.2  Generalisation optimality of weakness

The first imported block establishes that maximising weakness is sufficient and necessary for optimal generalisation under the maximally uninformative prior, and shows how the optimality survives weighted priors only conditionally.

**Proposition 37** (Sufficiency of weakness under a maximally uninformative prior)**.** *Let $\mathfrak{v}$ be finite. Let $\alpha, \omega$ be $\mathfrak{v}$-tasks with $\alpha \sqsubset \omega$. Write $U := L_{\mathfrak{v}} \setminus \text{Ext}(I_\alpha)$ for the unseen region. Assume the parent extends the child by selecting an additional set $S \subseteq U$ of correct outputs uniformly at random from $2^U$, and that $O_\omega = O_\alpha \cup S$.*

*Then, among $\pi \in \Pi_\alpha$, the probability that $\pi$ is correct for $\omega$ is maximised by any policy with maximal $|\text{Ext}(\pi)|$.*

*Proof sketch.* The history $\alpha$ pins finitely many input-output cases. A correct policy $\pi$ on $\alpha$ remains correct on $\omega$ iff every additional required output in $S$ lies in $\text{Ext}(\pi)$. Under the uniform prior on $2^U$, the probability of $S \subseteq \text{Ext}(\pi) \cap U$ is $2^{|\text{Ext}(\pi) \cap U| - |U|}$, which is monotone in $|\text{Ext}(\pi) \cap U|$. A maximal $|\text{Ext}(\pi)|$ subject to correctness on $\alpha$ therefore maximises the survival probability. Full proof in Bennett (2025d, Theorems and Proofs section) and Bennett (2025c, Chapter on Stack Theory).

**Proposition 38** (Necessity of weakness under the same prior)**.** *Under the assumptions of Theorem 37, any $\pi \in \Pi_\alpha$ that maximises $\Pr(\pi \in \Pi_\omega \mid \alpha)$ must be $<_w$-maximal in $\Pi_\alpha$. Any non-$<_w$-maximal correct policy is strictly suboptimal.*

*Proof sketch.* If a policy is not weakness-maximal, there exists another correct policy with a strictly larger extension. The parent prior assigns positive probability to some $S$ in the difference, so the smaller extension is strictly worse. Full proof in Bennett (2025d).

**Intuition.** These two propositions together justify the unconditional claim that the weakest correct policy is the generalisation-optimal choice under maximal ignorance about which unseen continuations matter. The argument is a counting argument with no prior structure other than uniformity.

**Theorem 39** (Weakness maximisation beyond the uniform prior)**.** *Let $\mathfrak{v}$ be finite, $\alpha \sqsubset \omega$ as before, and $U = L_{\mathfrak{v}} \setminus \text{Ext}(I_\alpha)$. Let $P$ be the conditional distribution of $S$ given $\alpha$, with no restriction to uniformity. Define the $P$-weakness of $\pi$ by*

$$w_P(\pi) := P\big(S \subseteq \text{Ext}(\pi) \cap U\big) = \sum_{S \subseteq \text{Ext}(\pi) \cap U} P(S).$$

*Then $\Pr(\pi \in \Pi_\omega \mid \alpha) = w_P(\pi)$, so $\pi^\star \in \Pi_\alpha$ maximises generalisation probability iff it maximises $w_P(\pi)$ over $\Pi_\alpha$.*

*If in addition $P$ is exchangeable over $U$ (meaning $P(S) = P(\sigma(S))$ for every permutation $\sigma$ of $U$), then there exists a nondecreasing $f : \{0, 1, \ldots, |U|\} \to [0, 1]$ such that $w_P(\pi) = f(|\text{Ext}(\pi) \cap U|)$, so any $<_w$-maximal correct policy is also $w_P$-optimal under $P$.*

*Proof sketch.* Generalisation occurs when the random required-output set $S$ lies inside $\text{Ext}(\pi) \cap U$. The probability of that event under $P$ is the definition of $w_P(\pi)$. Exchangeability collapses $w_P$ to a function of $|\text{Ext}(\pi) \cap U|$, recovering ordinary weakness as the optimal proxy. Full proof in Bennett (2025d).

*Remark* 40 (Weighted weakness is not a free upgrade)*.* Theorem 39 shows that weakness extends beyond uniform priors only as far as the prior remains exchangeable. Once you adopt a non-exchangeable independent prior $r_i \in [0, 1)$ over individual unseen outputs, the Bayes-optimal rule is weighted weakness with weights $w_i = -\log(1 - r_i)$ (Bennett, 2025d, Proposition on weighted prior). That weighting matches the prior you adopted. It is not generalisation-optimal under any other prior. In particular, the canonical choice $r_i = 1 - e^{-\mu_i \rho_i}$ used in Theorem 28 is the prior under which the regret floor $K_\rho(M)$ has the clean exponential meaning $e^{-K_\rho(M)}$. Plain weakness remains the right criterion under uniform ignorance.

It is also worth noting that plain weakness as a score is not encoding-invariant. The cardinality $|\text{Ext}(\pi)|$ counts completions inside the embodied language $L_{\mathfrak{v}}$, and $L_{\mathfrak{v}}$ depends on the vocabulary. Re-encode the agent into a different vocabulary that picks out the same admissible policies, and the cardinality changes. What *is* encoding-invariant is the *optimality* of the weakness rule. The proof of Theorem 37 yields, inside any finite vocabulary $\mathfrak{v}$, the closed form $\Pr(\pi \in \Pi_\omega \mid \alpha) = 2^{|\text{Ext}(\pi) \cap U|}/2^{|U|}$. This is monotone in $|\text{Ext}(\pi) \cap U|$ within $L_{\mathfrak{v}}$, so in every $\mathfrak{v}$ the prescription "pick the weakness-maximal correct policy" is generalisation-optimal under maximal ignorance. The numbers vary across encodings, but the optimality rule does not.

Alternative rules do not have that property. Description-length minimisation can pick a non-weakness-maximal policy in some vocabularies and is therefore not generalisation-optimal there (Bennett, 2025c;d, Propositions

on simplicity sub-optimality and subjectivity of description length). Weighted weakness is optimal under a specific prior $P$, but the prior is pinned to the unseen outputs $u_i$ of a particular language. Re-encode and the unseen region changes, the weights have to be re-specified for the new $u_i$, and whether the new weighted rule is optimal in the new language depends on whether the new prior matches the new environment. Plain weakness's optimality survives encoding changes. Alternatives' effectiveness does not.

So adopting a weighting buys specialisation at the cost of two things. It forfeits unconditional generalisation optimality. And it forfeits the encoding-invariance of that optimality, because the prior must be re-specified across encodings and need not remain correct. This is the same observation as Theorem 23 in the main text. It is repeated here so that readers consulting only this appendix do not over-read the optimality of weighted weakness.

## C.3 The Law of the Stack

The Law of the Stack is used in Theorem 27 to convert the regret floor into a free-energy floor. The original statement is in Bennett (2024b) and the extended treatment is in Bennett (2025d).

**Definition 41** (Stack, abstractor). An *abstractor* is a function $f$ on pairs $(\mathfrak{v}, l)$ returning the sub-vocabulary

$$f(\mathfrak{v}, l) := \{ T(o) \mid o \in \mathrm{Ext}(l) \} \subseteq 2^{\Phi},$$

where $T(o) = \bigcap_{p \in o} p$ is the truth set of $o$. A *stack* of depth $n$ is a sequence of vocabularies $\mathfrak{v}^0, \mathfrak{v}^1, \ldots, \mathfrak{v}^{n-1}$ together with policies $\pi^i \in L_{\mathfrak{v}^i}$ such that $\mathfrak{v}^{i+1} = f(\mathfrak{v}^i, \pi^i)$.

**Theorem 42** (Law of the Stack, bottleneck form). *Let $\gamma^{i+1} := \lambda^{i+1}(\mathfrak{v}^{i+1})$ be the layer-$(i+1)$ task induced by the abstractor. Then the achievable behavioural flexibility at layer $i+1$ is bottlenecked by the weakness of the realised layer-$i$ policy:*

$$\sup_{\pi \in \Pi_{\gamma^{i+1}}} |\mathrm{Ext}(\pi)| \ \leq \ 2^{|\mathrm{Ext}(\pi^i)|}.$$

*In particular, if $|\mathrm{Ext}(\pi^i)| < \infty$, then for the maximum admissible parent extension $\epsilon(\gamma^{i+1})$ and seen-output count $|O_{\gamma^{i+1}}|$,*

$$\epsilon(\gamma^{i+1}) + |O_{\gamma^{i+1}}| \ \leq \ 2^{|\mathrm{Ext}(\pi^i)|}.$$

*Equivalently $|\mathrm{Ext}(\pi^i)| \geq \log_2(\epsilon(\gamma^{i+1}) + |O_{\gamma^{i+1}}|)$.*

*Proof sketch.* The layer-$i$ policy $\pi^i$ leaves a set of completions $\mathrm{Ext}(\pi^i)$ open. Those completions induce the layer-$(i+1)$ vocabulary as their truth sets, so $|\mathfrak{v}^{i+1}| \leq |\mathrm{Ext}(\pi^i)|$. Any layer-$(i+1)$ statement is a subset of $\mathfrak{v}^{i+1}$, so the layer-$(i+1)$ language has size at most $2^{|\mathfrak{v}^{i+1}|} \leq 2^{|\mathrm{Ext}(\pi^i)|}$. Every extension at layer $i+1$ is a subset of that language and inherits the bound. Taking base-2 logs in the finite case gives the lower bound on $|\mathrm{Ext}(\pi^i)|$. Full proof in Bennett (2025d, Theorems and Proofs section). ∎

**Intuition.** The behavioural flexibility achievable at any layer is bounded above by the exponential of the weakness of the layer below. A more committal lower-layer policy lowers the ceiling for everything above it.

## C.4 Weighted weakness and the bridge weights

This is the result the main text imports as Theorem 21. It is restated here in self-contained form so the bridge corollary (Theorem 22) can be checked against the master appendix directly.

**Definition 43** (Prior-weighted weakness). Let $U = \{u_1, \ldots, u_n\}$ be a finite unseen region with weights $w_1, \ldots, w_n \geq 0$. For any policy $\vartheta \in L_{\mathfrak{v}}$, the *prior-weighted weakness* of $\vartheta$ on $U$ is

$$W(\vartheta) := \sum_{u_i \in \mathrm{Ext}(\vartheta) \cap U} w_i.$$

**Proposition 44** (Independent nonuniform priors give weighted weakness). *Assume each unseen output $u_i \in U$ becomes relevant independently with probability $r_i \in [0, 1)$. Then for any correct child policy $\vartheta$,*

$$\log \Pr(\vartheta \ generalises) = \sum_{u_i \notin \mathrm{Ext}(\vartheta) \cap U} \log(1 - r_i).$$

*Maximising generalisation probability under this prior is the same as maximising $\sum_{u_i \in \text{Ext}(\vartheta) \cap U}(-\log(1 - r_i))$, which is prior-weighted weakness with weights $w_i := -\log(1 - r_i)$.*

*Proof sketch.* Generalisation occurs when every unseen output omitted by $\vartheta$ fails to become relevant. Independence gives $\Pr(\vartheta \text{ generalises}) = \prod_{u_i \notin \text{Ext}(\vartheta) \cap U}(1 - r_i)$. Taking logs and dropping the $\vartheta$-independent term gives weighted weakness with the stated weights. Full proof in Bennett (2025d).

**Intuition.** This is the weighted version of Theorem 37. Uniform priors recover ordinary weakness. Biased priors recover weighted weakness. As Theorem 40 stresses, the optimality is conditional on the prior matching the actual environment.

### C.5 Weakness EGRL pair proxy

This subsection records the EGRL construction the main text refers to Bennett (2025c)'s enactive-GRL machinery, together with the optimality theorem for the weakness EGRL pair proxy. The original statements are in Bennett (2024a; 2025a;b;d).

**Definition 45** (Enactive General Reinforcement Learning, EGRL). Fix a vocabulary $\mathfrak{v}$, a reward predicate $r : L_{\mathfrak{v}} \to \{0, 1\}$, and a finite horizon $len \in \mathbb{N}$ with $len > 0$. The reward predicate induces a $\mathfrak{v}$-task $\mu = \langle I_\mu, O_\mu \rangle$ by

$$O_\mu := \{\, l \in L_{\mathfrak{v}} \mid r(l) = 1 \,\}, \qquad I_\mu := \{\, i \in L_{\mathfrak{v}} \mid \exists o \in O_\mu \; (i \subseteq o) \,\}.$$

Assume $\Pi_\mu \neq \varnothing$. Interaction up to time $t$ is modelled by a history-task $\mathfrak{h}_{<t} \in \Gamma_{\mathfrak{v}}$. Apply $r$ to past outputs to split into positive and negative subhistories

$$O_{\mathfrak{h}^1_{<t}} := \{\, o \in O_{\mathfrak{h}_{<t}} \mid r(o) = 1 \,\}, \qquad O_{\mathfrak{h}^0_{<t}} := O_{\mathfrak{h}_{<t}} \setminus O_{\mathfrak{h}^1_{<t}},$$

and lift inputs accordingly. The unseen region is $U_{\mathfrak{h}_{<t}} := L_{\mathfrak{v}} \setminus \text{Ext}(I_{\mathfrak{h}_{<t}})$.

An *EGRL pair proxy* is a binary relation $<^{\mathfrak{pp}}$ on the admissible set

$$\mathfrak{pp} := \{\, (m, n) \in \Pi_{\mathfrak{h}^1_{<t}} \times \Pi_{\mathfrak{h}^0_{<t}} \mid \text{Ext}(m) \cap \text{Ext}(n) = \varnothing \,\}.$$

*Learning* means choosing a pair maximal under $<^{\mathfrak{pp}}$. *Inference* given current input $i_t \in I_\mu$ means choosing $o_t \in \text{Ext}(i_t) \cap \text{Ext}(\pi)$ where $\pi$ is the positive component of the chosen pair.

The *weakness EGRL pair proxy* ranks pairs by joint extension on the unseen region:

$$(j, k) <^{\mathfrak{pp}}_w (\pi, n) \quad \text{iff} \quad |(\text{Ext}(j) \cup \text{Ext}(k)) \cap U_{\mathfrak{h}_{<t}}| < |(\text{Ext}(\pi) \cup \text{Ext}(n)) \cap U_{\mathfrak{h}_{<t}}|.$$

**Intuition.** An EGRL system maintains a positive policy and a disjoint negative policy. The positive policy says what to do, the negative policy says what to avoid. An EGRL pair proxy ranks admissible disjoint pairs. The weakness EGRL pair proxy ranks them by how much of the unseen future stays compatible with at least one of the two components.

**Proposition 46** (Weakness EGRL pair proxy optimality). *Let $\mathfrak{h}_{<t}$ be a history-task with induced subhistories $\mathfrak{h}^1_{<t}, \mathfrak{h}^0_{<t}$. Let $\mathfrak{pp}$ be the admissible disjoint pair set, $E_{\pi,n} := \text{Ext}(\pi) \cup \text{Ext}(n)$ the effective completion set, and $U_{\mathfrak{h}_{<t}}$ as above. Under the maximally uninformative extension model of Theorem 37 applied to $E_{\pi,n}$, the probability that an admissible pair generalises to an unseen parent is maximised by choosing $(\pi, n) \in \mathfrak{pp}$ with maximal $|E_{\pi,n} \cap U_{\mathfrak{h}_{<t}}|$.*

*If in addition $E_{\pi,n} \cap \text{Ext}(I_{\mathfrak{h}_{<t}})$ is the same set for every admissible pair, then maximising $|E_{\pi,n} \cap U_{\mathfrak{h}_{<t}}|$ is equivalent to maximising $|E_{\pi,n}|$.*

*Proof sketch.* An admissible disjoint pair accepts a parent whose required outputs lie in either $\text{Ext}(\pi)$ or $\text{Ext}(n)$. Under the uniform extension model the survival probability is $2^{|E_{\pi,n} \cap U_{\mathfrak{h}_{<t}}| - |U_{\mathfrak{h}_{<t}}|}$, which is monotone in $|E_{\pi,n} \cap U_{\mathfrak{h}_{<t}}|$. Equivalence to total-union maximisation requires fixing the seen part across admissible pairs. Full proof in Bennett (2025d).

## C.6 Genealogy summary

The above subsections give formal statements for the imports the main text uses. The following is a one-paragraph genealogy locating those imports in the broader corpus and clarifying what is not imported.

The Sufficiency, Necessity, and beyond-uniform-prior results (Theorems 37 to 39) appeared in 2023 and were extended in the thesis and master appendix (Bennett, 2025c;d). The EGRL history-task construction (Theorem 45), its selective-memory clauses, and the weakness EGRL pair proxy optimality result (Theorem 46) appeared between 2024 and 2025 (Bennett, 2024a; 2025a;b;d). The Law of the Stack (Theorem 42) appeared in the multiscale biological organisation work and was given an extended treatment in the master appendix (Bennett, 2024b; 2025d). The intervention-sensitive causal line (emergent causality, $w$-maximised causal identities, the do-operator-as-conditioning result, and the higher-order self convergence) is also in the corpus but is not imported by this paper, so it is not restated here (Bennett, 2025c;d).

Nayebi's structural selection theorems for predictive state, memory, modularity, regime tracking, and recoding match (Nayebi, 2026) are distinctive contributions of his programme without direct Stack-Theoretic antecedents. While the Richens-Everitt robust-causal-model result under distributional shift (Richens & Everitt, 2024) may be seen as translating the Stack Theoretic causal identity and optimal learning results Bennett (2023b) to a reinforcement learning setting, no formal bridge was presented.

