# OpenReview forum: "Regret Is Weighted Forgetting"
_TMLR — Rejected by TMLR_

### Review · Reviewer_Yrx5 · 2026-04-22

**Summary Of Contributions:**

This paper proves an exact identity (Theorem 7): for a fixed representation M, the minimum average normalised regret over all M-based policies equals the minimum margin-weighted deletion cost K_ρ(M) needed to make the optimal bet single-valued on each representation-test cell. The same quantity is reformulated as a weighted weakness deficit in Stack-Theoretic language (Corollary 18). From this identity the authors derive a policy-wise regret decomposition, a regret-based partial order on abstractions, a representation-convergence theorem, a Lipschitz stability result, and cross-framework corollaries linking the regret floor to free energy, a generalisation probability e^(−K_ρ(M)), and multi-agent coordination cost. Controlled POMDP experiments verify the identity through two independent code paths and test K_ρ as a representation diagnostic.

Key strengths include a clean main identity with a carefully chosen margin weighting that makes the bridge exact rather than up-to-constants, and explicit scope management about what is new, what is imported, and what is open. Key weaknesses involve heavy reliance of the cross-framework corollaries on a prior corpus unfamiliar to the typical TMLR reader, and experimental scale that is small relative to the practical claims made for K_ρ as a diagnostic.

**Additional Comments:**

This paper builds heavily on Bennett's Stack-Theoretic corpus (Bennett 2023a,b; 2024a,b; 2025a,b,c,d) and on Nayebi's 2026 selection theorems, neither of which falls within my primary area of expertise. Many of the core objects of the paper, including lifted diagnostic tasks, weighted weakness, the Law of the Stack, admissible policies in an embodied language, and the surrounding genealogy, required substantial effort to approach within the allotted review time, and within that time I was not able to verify them end-to-end to the level I would normally apply.

**Audience:**

Yes

**Audience Explanation:**

The state-abstraction and bisimulation communities would find the regret-based partial order and the representation-convergence theorem directly relevant, as they complement rather than duplicate existing bisimulation-metric tools. Researchers working on representation diagnostics for deep RL would find K_ρ potentially useful as a training-time monitor that makes the representational-to-policy-bottleneck transition quantitatively visible. The generalisation-probability corollary e^(−K_ρ(M)) gives a predictive semantics that bridges a regret floor and a probability bound through an exact identity rather than an analogy.

**Claims And Evidence:**

Yes

**Claims Explanation:**

Assuming the foundational Stack-Theoretic background and the cited lemmas imported from prior work are accurate as stated, the core claims of this paper are logically self-consistent and supported by sufficient evidence.

1

The core identity (Theorem 7) and its immediate consequences stated entirely in RL language have short, transparent proofs that I was able to follow and partially reproduce. Both sides of the identity reduce cellwise to min{A_C, B_C}, the policy-wise decomposition follows from elementary algebra on an affine function, and the multi-class generalisation and measurable version are natural extensions of the same argument. The derived corollaries on regret-based partial order, representation convergence, and Lipschitz stability rest only on Theorem 7 and basic partition combinatorics, and their reasoning appears correct to the extent that I was able to check.

2

The cross-framework corollaries in Section 4 and Section 5.2 depend on results imported from an extensive prior corpus in Stack Theory. The authors flag these external dependencies explicitly. Conditional on the imported results being correct in their cited forms, the derivations in this paper are sound. The two-sided POMDP verification, in which a Stack-Theoretic truth-set computation and a direct RL cellwise computation agree exactly across 600 independent checks, offers methodological reassurance that the bridge is not a post-hoc relabelling.

**Requested Changes:**

It is recommended that the authors extract the specific Stack-Theoretic results imported from prior work, at minimum the Law of the Stack used in Corollary 22 and the weighted-weakness or weakness-maximisation results underlying Proposition 17 and Corollary 18, and state them as self-contained propositions with full statements and precise conditions in an appendix. Appendix C at present lists names of antecedents rather than their formal content, which makes the cross-framework corollaries hard to assess for readers not already embedded in that literature. This change would substantially improve the paper's accessibility and verifiability for the TMLR audience.

---

> ### Author Response · Authors · 2026-05-07
>
> Thank you for the constructive review and for identifying the core identity as a useful contribution. I took your main recommendation to be that the paper should separate the RL theorem from the Stack-Theoretic bridge, and that the imported Stack-Theoretic machinery should be made a bit more self-contained, rather than relying so much on references.
>
> I have revised the manuscript accordingly. Histories, tests, binary test outcomes, \(p_T(h)\), margin weights, the finite evaluation distribution, representation-test cells, and \(M\)-based policies are all defined before \(K_\rho(M)\) is introduced. The main theorem result is then stated as a weighted deletion-cost identity for the quotient induced by \((M(h),T)\), with the weighted Bayes-error reading made explicit in the same section.
>
> The formal Stack-Theoretic bridge now comes after the RL setup and exact identity, which changes the order of dependency. A reader can now evaluate the regret theorem without needing to know Stack Theory, and then read the bridge as an interpretation and extension of the same quantity.
>
> I also expanded the Stack-Theoretic appendix. The revision states the relevant notions of embodied language, extension, correct policy, weakness, weighted weakness, the Law of the Stack, and EGRL directly in the paper. I also clarified the relation between weakness and encoding. Weakness is intrinsic relative to a fixed embodied vocabulary, but the raw numerical weakness score is not asserted to be invariant across vocabularies. The invariant claim concerns the optimality of the weakness rule inside a fixed embodied language.
>
> I also clarified the free-energy result. The corollary now states the weighted lifted-language assumption under which the Law of the Stack is being applied. This avoids treating an ordinary extension cardinality and a weighted retained mass as if they were automatically the same object.
>
> I moderated the interpretation of the experiments as well. The experiments are controlled demonstrations of the decomposition and of the diagnostic behaviour of \(K_\rho\). They are not presented as a large-scale practical claim about representation learning. The Boolean experiment now states that the support points are partial observation histories, with probabilities obtained by marginalising over hidden full Boolean states. The neural-network discussion now separates end-to-end classifier accuracy from the irreducible aliasing cost measured by the discretised representation-test quotient.
>
> The Stack-Theoretic material now functions as a bridge from a regret identity to the broader weakness framework, rather than as a prerequisite for understanding the identity.

---

### Review · Reviewer_1nD6 · 2026-04-24

**Summary Of Contributions:**

The authors conduct a theory-based analysis of representation versus policy for the regret of RL agents. They study this through the lens of the Stack-Theoretic framework.

Unfortunately, I was not able to follow this paper closely. It is not sufficiently self-contained for a professional reader to understand, particularly if the reader is unfamiliar with this specific area of the RL literature. For example, in the first paragraphs of the introduction, the reader is assumed to be familiar with the following terminology, without explanations or definitions:
- representation cell
- quotient history-test pairs
- the Stack-Theoretic programme
- natural defect variables
- EGRL (acronym undefined)

These are just a few examples rather than an exhaustive list, and similar issues are present in the subsequent sections.

In Section 3, much of the notation is either undefined or not clearly and fully defined. For example, x_i is defined as (h_i, T_i) without formally defining what a history h or a test T is. The notation p_{T_i}(h_i) appears to be completely undefined. Again, these are just a few examples rather than an exhaustive list of the problems.

Minor quibble: The author uses the first-person “I” rather than the traditional academic “we”. Some may find this to be inappropriate for a venue such as TMLR.

**Audience:**

No

**Audience Explanation:**

Unfortunately, because of the issues above, I cannot evaluate the significance and potential interest.

**Claims And Evidence:**

No

**Claims Explanation:**

Unfortunately, because of the issues above, I cannot evaluate the claims and evidence in detail.

**Requested Changes:**

I found the paper in its current form to be completely inaccessible to a reader without in-depth knowledge of this narrow area of the literature. Major changes throughout the paper are likely needed to address these issues.

---

> ### Author Response · Authors · 2026-05-07
>
> Thank you for the review. I read your main concern as being about self-containedness. In particular, the original version introduced several objects before giving the reader enough local definitions to evaluate the theorem. I agree that this made the paper harder to assess than it needed to be and have taken corrective action.
>
> The revision is designed so that the main RL identity can be evaluated without prior familiarity with Stack Theory. I now define the finite evaluation support, histories \(h\), tests \(T\), binary outcomes \(Z_T\), the conditional probabilities \(p_T(h)\), optimal labels \(y_i\), margin weights \(\rho_i\), representation-test cells, and \(M\)-based policies before introducing \(K_\rho(M)\). The phrase “representation-test cell” now means a cell of the quotient induced by \((M(h),T)\). In other words, two support points are in the same cell when the representation gives the same state and the same diagnostic query is being asked.
>
> I have also added an explicit scope note about regret. The paper studies fixed-distribution normalised betting regret over history-test pairs. It does not claim to be proving a theorem about cumulative regret in online RL, value loss in arbitrary MDPs, or worst-case minimax regret. The narrower scope is deliberate, because it permits an exact cell-by-cell decomposition.
>
> I also added a fuller worked example before the main theorem. The example walks through the representation-test partition and shows how a mixed cell contributes the cheaper of its two weighted label masses to \(K_\rho(M)\). The goal is that a reader can understand the calculation before reading the theorem.
>
> The Stack-Theoretic terminology has also been reorganised. In the original version, terms like weighted forgetting, weakness deficit, EGRL, selective forgetting, and pair-proxy machinery appeared before enough local context had been established. In the revision, the RL setup and theorem are stated first in self-contained terms, and the formal Stack-Theoretic bridge is developed later. The appendix has been expanded so the relevant Stack-Theoretic definitions are present in the manuscript itself rather than assumed from earlier papers.
>
> I also clarified the provenance of the Stack-Theoretic side. The EGRL programme, selective-memory clauses, and pair-proxy machinery are not claimed as new contributions of this paper. The new contribution is the exact identification between the fixed-distribution representation-regret floor and the weighted deletion or weakness-deficit quantity.
>
> The main theorem should now be readable as a standalone statement. Given a representation \(M\), quotient the finite evaluation support by \((M(h),T)\). In each cell, a policy that only sees \(M(h)\) and \(T\) must make one report for the whole cell. If the cell contains support points with different optimal labels, some regret is unavoidable. The theorem states that the minimum such regret is exactly the sum over cells of the cheaper weighted label mass. The policy-wise decomposition then says that any actual policy’s regret equals this irreducible representation cost plus avoidable within-cell misreporting.

---

### Review · Reviewer_DWND · 2026-04-27

**Summary Of Contributions:**

This paper studies how much regret is forced by a fixed representation. The main result shows that, for a fixed representation (M) and a finite evaluation distribution over history-test pairs, the minimum normalized regret over all policies using that representation is equal to a margin-weighted deletion cost. Intuitively, once histories are grouped into representation-test cells, regret comes from cells where the same representation still contains examples with different optimal bets. The paper then interprets this quantity as weighted forgetting and connects it to Stack-Theoretic weakness, generalization probability, free energy, and multi-agent coordination.

I found the core Section 3 identity interesting and potentially useful as a representation diagnostic. However, the current presentation is hard to follow. The paper tries to connect many frameworks at once, which makes it difficult to separate the main technical contribution from broader interpretations.

### Strengths:

- The main decomposition in Section 3 is clean. It gives a useful way to separate regret caused by the representation from regret caused by a suboptimal policy on top of that representation.
- The idea of measuring representation quality through the weighted impurity of representation-test cells is intuitive once the setup is understood.
- The experiments are small, but they help illustrate that ($K_\rho(M)$) can behave differently from raw impurity or accuracy.

### Weaknesses:

- The paper is difficult to read because many basic objects are introduced too quickly. In particular, the definitions of (h), (T), and (p_T(h)) are mostly assumed from prior work. This makes the setup hard to understand for readers who are not already familiar with the betting-test framework.
- The “test” abstraction is not formalized enough. Since the quotient cells are defined using both (M(h)) and (T), the paper should explain what information the policy receives and what kind of object a test is.
- The paper would benefit from a much smoother build-up. The main identity can be explained as weighted classification error, or weighted Bayes error, under a partition induced by the representation-test cells. Instead, the presentation quickly moves from RL regret to weighted forgetting, Stack-Theoretic weakness, free energy, generalization probability, and multi-agent coordination. This makes the contribution feel broader and less focused than what is directly proved.
- The regret notion is narrower than standard RL regret. It is a fixed-distribution normalized betting regret, not cumulative regret or standard value loss in an MDP. This is fine, but the scope should be made clear in the introduction and in the framing of the paper.

**Audience:**

Yes

**Audience Explanation:**

I think the topic is potentially interesting to the TMLR community. Representation quality in partially observable settings is an important problem, and a diagnostic that separates representation-induced regret from policy-induced regret could be useful.

That said, the current paper may lose many readers because of the amount of background assumed and the number of frameworks introduced. A clearer version that foregrounds the RL abstraction result and then presents the Stack-Theoretic connection as a secondary interpretation would likely have broader appeal.

**Broader Impact Concerns:**

I do not see major ethical concerns. The work is mostly theoretical and uses controlled diagnostic experiments. The main risk is overstatement of the scope of the results, especially if readers interpret the regret identity as applying more broadly to RL than it currently does.

**Claims And Evidence:**

Yes

**Claims Explanation:**

I think the main Section 3 claims are supported, assuming the setup is interpreted as a fixed diagnostic distribution with known conditional probabilities. The algebraic decomposition appears sound and the experiments are consistent with it.

However, I am less convinced by the broader framing. The Stack-Theoretic, free-energy, generalization-probability, and multi-agent claims seem more assumption-dependent than the core identity. I would prefer the paper to clearly separate the direct theorem from these later interpretations.

**Requested Changes:**

1. **Critical:** Define the basic setup more carefully. Please give clear definitions and examples for (h), (T), ($p_T(h)$), the evaluation distribution, and the policy’s information. State whether $p_T(h)$ is the true environment probability, an empirical estimate, or the agent’s belief. If the theorem assumes the true probability, please say so explicitly.

2. **Critical:** Clarify the scope of the regret notion. The paper should clearly distinguish fixed-distribution normalized betting regret from standard cumulative RL regret or MDP value loss.

3. **Critical:** Explain the relation between ($K_\rho(M)$) and weighted Bayes error under a partition much earlier. This would make the main result easier to understand and would also clarify the novelty.

4. **Would strengthen the paper:** Rename or carefully explain “weighted forgetting.” The current term may mislead readers into thinking of continual learning or neural forgetting, while the paper means deletion of support points to purify cells.

5. **Would strengthen the paper:** Add a simple running example before the main theorem. A small example with a few histories, tests, representation cells, and labels would make the result much easier to follow.

6. **Would strengthen the paper:** Reduce the breadth of the framing. The paper currently tries to connect RL, weighted forgetting, Stack Theory, free energy, generalization, causality, and multi-agent coordination. A more focused version would be more convincing.

---

> ### Author Response · Authors · 2026-05-07
>
> Thank you for the detailed and useful review. I read your central point as follows. The main identity is clean, but the original presentation made the reader work too hard before the objects in that identity were fully defined. I have revised the manuscript around that point.
>
> The finite diagnostic setup now comes before the main theorem. A test \(T\) is defined as a diagnostic query, its binary outcome is written \(Z_T\in\{0,1\}\), and \(p_T(h)=\Pr(Z_T=1\mid h)\) is introduced as the true environment probability that the test fires after history \(h\). This also clarifies that the equivalence condition \(T_i=T_j\) means the same diagnostic query is being applied, not that the same realised outcome was observed.
>
> I have also made the regret scope explicit near the start of the paper. The result concerns fixed-distribution normalised betting regret over a finite evaluation support of history-test pairs. It is not cumulative online regret, MDP value loss, or worst-case minimax regret. This scope restriction is important for the cellwise identity.
>
> I also took seriously your point that the quantity should be presented in standard representation-learning language before the Stack-Theoretic bridge is developed. In the revision, \(K_\rho(M)\) is introduced as the cellwise margin-weighted deletion cost induced by the representation-test partition \((M(h),T)\), with the weighted Bayes-error reading stated in the same section. The formal Stack-Theoretic bridge now comes after the RL setup and identity have been stated.
>
> I added a fuller worked example before the main theorem so that the reader can see the deletion calculation inside a mixed representation-test cell before encountering the abstract statement.
>
> I also kept the relation to state abstraction and bisimulation in standard RL language, so that the main identity is positioned relative to familiar abstraction tools before the Stack-Theoretic bridge. The result is complementary to bisimulation metrics. Bisimulation metrics bound value-function loss, while the theorem identifies the fixed-distribution regret cost of a given quotient.
>
> Finally, I reduced the burden placed on the Stack-Theoretic material in the main argument. The Stack-Theoretic bridge remains part of the contribution, but the revised version no longer requires a reader to understand Stack Theory in order to evaluate the RL theorem. I also expanded the appendix so that the imported notions of weakness, extensions, correct policy, weighted weakness, and the Law of the Stack are stated directly.
>
> The experiments have also been clarified. They are controlled checks of the exact decomposition and of \(K_\rho\) as a representation diagnostic. They are not presented as a claim that the current experiments solve large-scale representation learning.

---

### Decision · Action_Editor_ocwv · 2026-06-05

**Recommendation:** Reject

**Audience:**

Yes

**Audience Explanation:**

The paper makes connections other fields that the reviewers appreciate.

**Claims And Evidence:**

No

**Claims Explanation:**

Even after revision, the paper is still not sufficiently self-contained. It relies heavily on terminology that the audience cannot be assumed to be familiar with.

As one of the reviewers pointed out, this includes but is not exhaustive to
* representation cell
* quotient history-test pairs
* the Stack-Theoretic programme
* natural defect variables

As such, it is even difficult for the AE to verify the correctness of the results without significant reading of prior work.

**Resubmission Of Major Revision:**

The authors may consider submitting a major revision at a later time.